# Much More Than a Scaffold: Cytoskeletal Proteins in Neurological Disorders

**DOI:** 10.3390/cells9020358

**Published:** 2020-02-04

**Authors:** Diana C. Muñoz-Lasso, Carlos Romá-Mateo, Federico V. Pallardó, Pilar Gonzalez-Cabo

**Affiliations:** 1Department of Oncogenomics, Academic Medical Center, 1105 AZ Amsterdam, The Netherlands; d.c.munozlasso@amsterdamumc.nl; 2Department of Physiology, Faculty of Medicine and Dentistry. University of Valencia-INCLIVA, 46010 Valencia, Spain; Carlos.Roma@uv.es (C.R.-M.); federico.v.pallardo@uv.es (F.V.P.); 3CIBER de Enfermedades Raras (CIBERER), 46010 Valencia, Spain; 4Associated Unit for Rare Diseases INCLIVA-CIPF, 46010 Valencia, Spain

**Keywords:** actin, tubulin, neurofilaments, microtubules, neuron, growth cone, cytoskeleton, neurological diseases

## Abstract

Recent observations related to the structure of the cytoskeleton in neurons and novel cytoskeletal abnormalities involved in the pathophysiology of some neurological diseases are changing our view on the function of the cytoskeletal proteins in the nervous system. These efforts allow a better understanding of the molecular mechanisms underlying neurological diseases and allow us to see beyond our current knowledge for the development of new treatments. The neuronal cytoskeleton can be described as an organelle formed by the three-dimensional lattice of the three main families of filaments: actin filaments, microtubules, and neurofilaments. This organelle organizes well-defined structures within neurons (cell bodies and axons), which allow their proper development and function through life. Here, we will provide an overview of both the basic and novel concepts related to those cytoskeletal proteins, which are emerging as potential targets in the study of the pathophysiological mechanisms underlying neurological disorders.

## 1. Neuronal Cytoskeleton

The cytoskeleton is a cellular organelle formed by a three-dimensional scaffold of proteins that is particularly essential for the definition of the physiological properties of neurons. Since the development of the nervous system and throughout its entire existence, the cytoskeleton is essential to maintain neuronal functions. Neurons are morphologically distinguished from other cells because they have a compartmentalized structure: dendrites, a cell body (or neuronal soma), an axon, and axon terminals. In each compartment, cytoskeletal proteins have specific functions that ensure, as a final goal, the transmission of electrical and chemical signals between neurons. The neuronal cytoskeleton has to be both a flexible and dynamic organelle to maintain the neuronal circuit functioning through the life of an organism. It depends on three classes of filaments: intermediate filaments (IF), which are protein-based neurofilaments (10 nm in diameter), actin-based microfilaments (6 nm in diameter), and tubulin-based microtubules (24 nm in diameter) (Figure 1). In addition to these proteins, other filament-binding proteins regulate the organization and dynamics of these components.

In neurons, neurofilaments are the structural core of myelinated axons and modulate the axonal diameter [1,2,3], which is essential for maintaining axonal transport [4] and nerve conduction velocity [5,6]. In mature axons, microfilaments form both stable structures, such as actin rings, and dynamic structures, such as hotspots and trails. In developing or regenerating axons of all neurons, microfilaments form dynamic structures, such as the lamellipodia and filopodia, in the growth cones (GCs). On the other side, microtubules overlap to provide neurons with continuous transport tracks that allow active mitochondrial [7,8], vesicular [9,10], and mRNA [11,12] transport along the axons and in the growth cone, thus ensuring neuronal homeostasis.

These three elements (neurofilaments, microfilaments, and microtubules) work together to guarantee a proper formation of the nervous system during the embryonic development and to assure its function in adulthood. There are three particular events where the function of cytoskeletal proteins is required: first, during embryonic development, where the cytoskeleton participates in the growth and guidance of axons (for a review see, [13,14]); second, during adult life, where neurons depend on the cytoskeleton for maintaining neuronal homeostasis and neuronal plasticity [15,16,17,18]; and third, when the peripheral axon needs to regenerate after being injured. In this last event, peripheral neurons (for a review, see [19]) require a specific set of cytoskeletal proteins to ensure nerve regeneration upon damage [20,21].

Given the importance of cytoskeleton for neurons, it is not surprising that several neurological disorders either involve changes in the expression, dynamics, and stability of cytoskeletal proteins or are the result of their mutations. In this review, we will study the basic knowledge of the structure and function of cytoskeletal proteins in neurons, and highlight discoveries that may be relevant to understand the molecular mechanisms underlying some neurological disorders.

### 1.1. Intermediate Filaments

Intermediate filaments (IFs) are the major cytoskeletal proteins in neurons. In contrast to microfilaments and microtubules, IFs are more heterogeneous, have tensile strength, do not have polarity, and do not participate in cell motility. There are seven classes of IFs-proteins (Table 1) and among these, three classes have a distinctive function in cells of the nervous system (neurons and glial cells): Type III (Desmin, DES; the Glial Fibrillary Acidic Protein, GFAP; Peripherin, PRPH and Vimentin, VIM); Type IV (α-Internexin, INA; Neurofilament light, NF-L; Neurofilament medium, NF-M; Neurofilament heavy, NF-H and Syncoilin, SYNC); and Type VI (Nestin, NES and Synemin, SYNM).

#### 1.1.1. Intermediate Filaments and Their Assembly in Neurons

At the structural level, all the IFs-proteins are alpha-helical monomers with a head domain (amino-terminal end), a central rod region (alpha-helical region), and a tail domain (carboxy-terminal end) (Figure 2). The stage of the development of the nervous system determines which IF proteins will be used to build the neurofilaments (NFs) [22,23]. At the early stages of neuronal development, Nestin (NES) and Vimentin (VIM) [24,25] are the major subunits of IFs. As neuronal differentiation continues, neuronal-specific subunits replace NES and VIM. Among these, NF-L (~68–70 kDa) is the first subunit to appear; later on, its expression overlaps with that of INA (~55 kDa) and PRPH (~54 kDa) [26,27,28]. As neurons become mature, they trigger the expression of NF-M (~145–160 kDa) and NF-H (~200–220 kDa). Therefore, the mature NFs are heteropolymers made of combinations of NF-L, NF-M, NF-H with INA in the central nervous system (CNS) or with PRPH in the peripheral nervous system (PNS) (Figure 3).

Recent studies have highlighted novel functions for other intermediate filament proteins within the cells of the nervous system, under normal and pathological conditions [30,31]. These proteins are Synemin (SYNM), Syncoilin (SYNC), and GFAP. SYNM and SYNC are part of the intermediate filaments in developing neurons from the CNS and the PNS [32,33]. On the other side, GFAP is the major intermediate filament protein in astrocytes and is abundantly expressed in Schwann cells, where VIM is also expressed [34,35].

#### 1.1.2. Post-Translational Modifications in Neurofilaments

Many post-translational modifications (PTMs) have been reported to change the chemical properties of NFs; although some of them, like nitration, oxidation [36], and ubiquitylation, are still not well understood [37], other well established PTMs, like phosphorylation and glycosylation [38], deserve further review.

NF proteins are known to be the most highly phosphorylated proteins in the brain tissue [39], which indicates the importance of this modification for the function of NFs in the nervous system. There is substantial evidence showing how the phosphorylation state of NF proteins determines the localization of the NF within neurons (neuronal soma, axon, dendrites or synapses). NF proteins are synthesized in the neuronal soma and then are translocated to axons and dendrites, where they undergo selective phosphorylation. It is known that the phosphorylation of the head domain as soon as they are synthesized [40,41,42] maintains the disassembled state of NF proteins in the neuronal soma [43,44,45]. Just before NF proteins are translocated to the axons, several previously added phosphates on the head domain are removed to allow the co-assembly of NF proteins [46]. While NFs are transported along the axon, the tail domains of the NF-M and NF-H are subject to extensive phosphorylation on Lysine-Serine-Proline repeats (KSP) [39,46,47,48] (Figure 2). As opposed to what occurs in axons, the tail domains of NF proteins remain non-phosphorylated in neuronal soma and dendrites [39,49].

In addition to its role as a signal for the assembly and organization of NFs within neurons, the phosphorylation of one particular NF subunit, NF-H, has been recently linked to the optimal maintenance of the internode length (the section of the axon between nodes of Ranvier) [50]. A correct internode length is essential to maintain the maximal conduction velocity of peripheral nerves [51,52]. The myelin-dependent phosphorylation of the KSP repeats within the C-terminal domain of NF-M is a key step in determining internode length [50]. The phosphorylation of NFs remains a focus of active research because several neurodegenerative disorders have been linked to the disruption of the phosphorylation state of NFs [39] in Alzheimer’s Disease [53], Parkinson’s Disease [54], Amyotrophic Lateral Sclerosis [55], Multiple Sclerosis [56], and Charcot Marie Tooth [57,58].

Regarding glycosylation, O-GlcNAcylation is one of the most abundant PTMs in cytoskeletal proteins, but it remains an overlooked modification specifically in NF proteins [59]. NF-L, NF-M, and NF-H are extensively glycosylated with O-linked β-*N*-acetylglucosamine (O-GlcNAc) at the hydroxyl groups of serine (Ser) or threonine (Thr) residues of their head domains [40,60]. Although the abundance of O-GlcNAcylation in mammalian cells indicates the importance of this PTM for the neuron [59,60,61], we are still starting to discover its functional consequences. Recently, the first proof that O-GlcNAcylation can modulate interactions between VIM monomers has been revealed [38]. The study of Tarbet et al. shows that it is required for vimentin-dependent cell migration and the replication of *Chlamydia* bacteria inside cells after infection. Besides this relevant new study, other reports support the role of O-GlcNAcylation in other functions, such as regulating protein phosphorylation, preventing protein degradation, adjusting the localization of proteins, modulating protein-protein interactions, and mediating transcription, (for a review see [62]).

### 1.2. Actin Filaments (Microfilaments)

Actin filaments (F-actin) are an abundant component of the axonal cytoskeleton and are made by the polymerization of globular-actin (G-actin) [63,64], a protein highly conserved in all eukaryotic cells (for a detailed review of the structure of actin, see [65]). There are three isoforms in mammalian organisms: α, β, and γ, but α and mostly β are expressed in neurons [66]. Actin exists as its monomeric state (G-actin), which polymerizes and forms the actin filaments (F-actin) [67].

The amount of G- and F-actin in the cells is proportionally related to the type of cell and their particular needs. Interestingly, actin in neurons comprises about 4–7% of protein expression [68], and it rises to 7–8% of cell protein during brain development [69,70]. F-actin builds superior structures, such as actin-bundles and networks, which maintain and stabilize the plasma membrane of the cell body and the axon (also known as axolemma). It also controls the position of the nuclei [71,72], the dynamics of organelles, such as mitochondria, as well as the endoplasmic reticulum–mitochondria interaction [73,74]. In contrast to the NFs, which are quite stable and have a low exchange rate, actin filaments are more dynamic and have higher exchange rates as a response to the physiological needs of neurons. This characteristic makes the actin cytoskeleton essential during the development of the nervous system and for the process to regenerate axons after an injury, which is an intrinsic property of the long nerve fibers. Actin networks also provide the ways for axonal transport, allowing actin-based motor proteins (myosins) to transport organelles in short distances (for an overview see [75]).

#### 1.2.1. Actin and Actin-Binding Proteins (ABPs)

A specialized group of ABPs controls the assembly and disassembly of F-actin, as well as the formation of bundles and networks and their association with other cell structures. These ABPs are, in turn, under the control of specific signaling pathways (for a recent review of the ABPs, see [76,77]). Some of these proteins stand out among the others because their functions have been shown to be essential for neuronal function. For example, gelsolin (with two isoforms, one secreted-86 kDa and other cytoplasmic-81 kDa), is a calcium-regulated severing protein recognized by its neuroprotective activity [78,79,80]. Drebrin (48 kDa) binds to F-actin in dendritic spines changing its properties [81]. Talin (270 kDa) is an adhesion protein, which allows neuronal growth cone anchorage to the extracellular matrix [82,83,84]. α-Actinin (102 kDa) [85] and Filamin (281 kDa) [86,87] are F-actin cross-linking proteins that mediate spine morphology and the assembly of the neural post-synaptic densities. Profilin-1 (15 kDa) and Thymosin β-4 (Tβ4, 4.9 kDa) are sequesters of adenosine triphosphate-actin (ATP-actin) monomers (G-actin) to make the chemical available when needed. In mammalian cells, most actin monomers are bound either to Tβ-4 or profilin. On the other side, three isoforms of tropomyosins (*TPM1*, *TPM3,* and *TPM4*, ~33kDa) regulate the interaction of F-actin with other ABPs. Tropomyosin binds to F-actin and sequesters G-actin throughout isoform-specific mechanisms, the results of which are particularly indispensable for neuronal development and function [88]. Cofilin (19 kDa) is a pH-sensitive F-actin depolymerising protein [89]. The complex ARP2/3 (a seven-subunit complex, Arp2 (44 kDa) and Arp3 (47 kDa)) [90,91] mediates the creation of new actin-filaments [76,92]. Both cofilin and ARP2/3 carry out two indispensable tasks to maintain actin filament dynamics within neurons, as we will see next.

#### 1.2.2. Formation and Remodeling of Actin Filaments

Actin polymerization takes place in three stages (Figure 4). The first stage is the formation of a small aggregate consisting of three monomers of actin (Nucleation phase). Then, the filaments grow by the reversible addition of monomers at both ends of the nucleus (Elongation phase), but one of them, the barbed end (also known as the plus end), grows five to ten times faster than the pointed end (also known as minus end) [93,94]. While an actin filament grows, the concentration of actin monomers decreases until reaching the balance between F-actin and G-actin (Stabilization phase). Normally, G-actin is coupled to ATP. Although actin polymerization does not require ATP, there is evidence that G-actin coupled to ATP polymerizes faster than G-actin coupled to adenosine diphosphate (ADP) [95].

After an actin filament reaches the stabilization phase, the length of the filament is adjusted by a continuous turnover of actin. This process, known as actin treadmilling, is regulated by specialized ABPs that control the rotary exchange of actin (Figure 5) in the filament. These ABPs are profilin and cofilin, and each one contributes to this process by a particular mechanism. Profilin is a small protein known to promote the polymerization of actin filaments by charging monomeric globular actin (G-actin) with ATP or by reducing the critical concentration of free actin in the presence of other ABP, namely Thymosin-β4 [96,97] (Figure 5). First, profilin binds to the G-actin coupled to adenosine diphosphate (ADP) [98], promoting the loss of ADP and allowing ATP to bind [99], which is more abundant than ADP in physiological conditions. Later, the profilin-ATP-G-actin complex can only be attached to the barbed end of the filament. In doing so, profilin adds the subunit and dissociates itself due to its low affinity for the barbed ends [100]. Therefore, profilin activity does not increase the rate of the filament assembly, but it provides new ATP-G-actin subunits by accelerating ADP dissociation from the ADP-actin monomers. As a result, most of all, the G-actin within the cell is coupled to ATP. In mammalian cells, actin monomers are usually bound either to profilin or Tβ4. Tβ4 binds to ATP-G-actin subunits, inhibiting the addition of new subunits to either the plus or minus end of the actin filaments [101]. When some free actin is used to polymerize the filament, more Tβ4-ATP-G-actin complexes are dissociated, releasing more ATP-G-actin units. Then, Tβ4 acts as a reservoir for the unpolymerized actin within the mammalian cells to release them when it is needed. The role of Tβ4 has proved to be essential for neurons given its function as an anti-inflammatory and neuroprotective peptide [102,103,104,105].

Cofilin is one of the three isoforms belonging to the actin-depolymerizing factor ADF/cofilin superfamily identified in mammals: ADF (also known as destrin); cofilin-1, the major ubiquitous isoform in non-muscle tissues; and cofilin-2, the major isoform in the differentiated muscle [106,107]. Cofilin (1 and 2) is more abundant in neurons than ADF [108] and collaborates with other ABPs to reorganize actin cytoskeleton. Cofilin binds specifically to F-actin, preferably to the subunits coupled to ADP (with slower dynamics), which are the initial subunits in the filament and are located to the minus end of the filament. The binding of cofilin to two monomers of actin induces a rotation of the actin filament that destabilizes and breaks the filament into small pieces. Then, cofilin generates many more minus ends and therefore causes an increase in its disassembly rate (Figure 5). The ADP-G-actin subunits released in this process are sequestered by profilin, which speeds the conversion to ATP-G-actin and later the addition to the barbed end. The increment in the assembly rates of actin filaments is the result of the synergy between cofilin and profilin.

The activity of cofilin is modified by the phosphorylation and dephosphorylation of a serine residue (Ser3) [109,110]. The regulator proteins of this modification have been subject to extensive research due to the importance of the balance between phosphorylated and non-phosphorylated cofilin to maintain axonal homeostasis and to ensure the proper function of the growth cone motility and neurite outgrowth in developing and mature neurons. Two serine protein kinases (Lin11, Isl-1, and Mec-3 Kinase 1 and 2; LIMK1 and 2), under the control of the Rho family of GTPases [111,112,113], are known to catalyze the phosphorylation of cofilin, leading to inactivation [114]. Both kinases seem to be regulated by different proteins within these pathways (for a review of LIMK1 and 2 and their regulatory pathways, see [115]). On the other side, several phosphatases catalyze the dephosphorylation of cofilin (activation) by the removal of the previously added phosphate group at its Ser3: Slingshot (SSH) phosphatases (in mammalian cells three genes codify for SSH proteins: SSH1, SSH2, SSH3) [116,117]; chronophin/PDXP (CIN), a general phosphatase [118]; and the serine/threonine phosphatases type 1 and type 2A (PP1 and PP2A). Among the LIM-kinases and cofilin-phosphatases, LIMK1 and Slingshot-1 Like protein (SSH-1L) show a preference for regulating the activity of cofilin in neurons [111,119]. Moreover, a regulatory mechanism links SSH-1L and LIMK1 activity. SSH1-L plays a dual role because it dephosphorylates both cofilin and LIMK1, and hence increases the activity of cofilin and actin filament turnover [120]. An opposite effect can be seen in the serine/threonine p21-activating kinase 4 (PAK4), which can phosphorylate LIMK1 and also inactivate SSH1-L [120]. Besides PAK4, the protein kinase D (PKD 1 and 2) also have been shown to inactivate the SSH1L-based dephosphorylation of cofilin [121,122,123]. On the contrary, Calcineurin, a Ca^2+^/calmodulin-dependent protein phosphatase, has been shown to activate it [124,125].

In the case of chronophin, a mammalian haloacid dehalogenase (HAD)-type phosphatase, there is little knowledge of the direct regulation of its activity. However, available evidence shows that chronophin has a role in regulating actin dynamics at the leading edge of motile cells, (for a recent review of the research related to chronoipin, see [126]) and, more interestingly, that it seems to respond to the physiological levels of ATP in the cell [127]. Under normal levels of ATP, chronophin remains in a complex with the heat shock protein 90 (Hsp90), and, under the depletion of ATP, it is released of the complex and is therefore free to dephosphorylate cofilin. Thus, altered ATP levels as a result of pathological conditions can lead to aberrant cofilin/actin structures through these phosphorylation events, as will be discussed in Section 3.

#### 1.2.3. Actin Filament Branch Formation

The actin filament branches are essential for the development of dendritic networks (almost exclusively composed by actin filaments). They are also required for the proper function of axons and synapses. Furthermore, the branching of actin allows the construction of key cellular structures, such as lamellipodia and the structurally related membrane ruffles [128], which are essential to building motile structures such as the neuronal growth cone [129] and the actin waves that migrate along the axonal shaft of developing neurons [130,131,132]. The actin filament branch formation starts with the creation of an initial core of actin in an existing actin filament (Nucleation) (Figure 6). In this process, the ARP2/3 complex, considered the key nucleator of actin branches and is typically composed of two actin-related proteins (ARP2 (44 kDa) and ARP3 (47 kDa)) and five more subunits (ARPC1, ARPC2, ARPC3, ARPC4, ARPC5) [133], binds first to a nucleating promotor factor (NPF) and later to a mother actin filament; once this ARP2/3 complex is active, it changes its conformation, allowing ARP2 and ARP3 to imitate the barbed end of actin, providing the template to assemble a new filament. At this stage, the NPF helps to add the two first actin subunits of the new branch. Finally, the NPF dissociates from the new branch and more actin monomers are added to its barbed end [134,135,136,137,138,139].

The first NPF described to activate the ARP2/3 complex was the Wiskott–Aldrich Syndrome protein (WASP); later on, a homologous WASP, characterized for being enriched in neural tissues, was discovered and named N-WASP [140]. Some alternative ARP2/3 complexes, together with novel NPFs have been described. For example, particular ARP2/3 complexes containing vinculin alone or together with α-actinin (both ABPs) have been found in focal adhesions [141]. Other ARP2/3 complexes contain isoforms such as ARP3β [142] and ARPC5B [143] (preferentially expressed in brain cells) or phosphorylation variants of the canonical ARP2/3 subunits (for a recent review of these alternative ARP2/3 complexes, see [144]). Regarding NPFs, there are five families: the WASP proteins (WASP and N-WASP); the WASP family verprolin homolog isoforms (WAVE1–WAVE3); the WASP homolog associated with actin, membranes and microtubules (WHAMM); the WASP and SCAR homolog (WASH); and the junction-mediating regulatory protein (JMY) [145]. There is a long line of evidence that shows that WASP proteins are required to activate the ARP2/3 complex during several stages of brain development; in particular, to develop and maintain dendritic spines and synapses (for a review of the function of these NPFs in neuronal development see [15,146]).

In addition to the ARP2/3 complex, other nucleating systems exist in eukaryotic cells. These proteins include formins, cordon blue (COBL), and spire (also known as SPIR). Formins are the general name designating fifteen families of proteins known to share the Formin Homology 2 domain (FH2), and they perform the nucleation and elongation in unbranched linear actin filaments (for a review with detailed information of Formins, see [147]). Formins are involved in the assembly of actin trails (dynamic actin polymers) in the axons of developing hippocampal neurons [148]. On the other side, COBL and SPIR also form an actin nucleus that serves as “a seed” to polymerize actin filaments, similar to the ARP2/3 complex but using different mechanisms. COBL forms a trimeric actin nucleus while SPIR forms a two-start helical filament nucleus as a result of the binding of actin to the four G-actin binding domains in the protein [149]. In general, it is thought that all these nucleating systems have specific functions to adjust the architecture of actin cytoskeleton within the different compartments of neurons (for a recent review with a more detailed description of these nucleating proteins and its function in the nervous system, see [76,92,150]). One example of the regulating mechanisms that may govern the nucleation proteins was found in fission yeast cells. In this mechanism, the Profilin-G-actin complex benefits formin over the ARP2/3 complex-based mediated actin assembly [151]. The authors propose that in higher eukaryotic cells, with multiple profilin isoforms, the expression of different isoforms at a particular time, place, and tailored to facilitate actin assembly for different F-actin networks should exist [151].

### 1.3. Microtubules and Their Dynamics within the Axon

Microtubules (MTs) are a key polymer for the structure and function of neurons. MTs form tubular structures along the axon and dendrites resembling navigation pathways that are used by MTs-based motor proteins, such as kinesins and dyneins to transport vesicles, lipids, and essential organelles. MTs, similar to actin filaments, are highly dynamic and also are generated by the polymerization of dimers made of isotypes of α or β tubulin [152] (Figure 7). Tubulin is a superfamily of proteins that includes other different forms of tubulin (γ, δ, ε, ζ, η, θ, ι, κ), but the polymerization of α/β dimers makes MTs in mammalian neurons with a neuronal-specific isoform of the β isotype: β-III [153].

In neurons, MT assembly starts at the centrosome, the microtubule-organizing center located in the soma (Figure 7). There, α and β isotypes of tubulin form α/β dimers that later polymerize to produce a radial arrangement of MTs with the positive end (+, plus end) towards the periphery (Figure 7). Katanin, a microtubule-associated protein with ATPase activity [154,155], severs MTs coming from the centrosome and releases small and dynamic MTs with different sizes, allowing their delivery to the axon or outgrowing neurites. After that, MTs are transported along the axon and positioned with the same polarity (minus end → plus end) by either kinesins or dyneins. While being transported, MTs can undergo cycles of polymerization (growth) and depolymerization (shrinking) at the same time at its plus end. This process is known as dynamic instability [156] (Figure 7), and several factors intervene in modulating the transition of MTs from the growth to shrinking (catastrophe) and from shrinking to growth (rescue). The first is the intrinsic GTPase activity of tubulin [157,158] and the other is due to a group of proteins known as microtubule-associated proteins (MAPs). MAPs bind to the full length of MTs modifying their structure, stability, and dynamics.

#### 1.3.1. Microtubule-Associated Proteins (MAPs)

MAPs are a group of proteins that modulate several functions of MTs. Among these, a few are known to play a key role in neurons. For example, the microtubule-associated protein-1A and 1B (MAP1A and MAP1B) [159] are essential to the development of central and peripheral neuronal networks and perhaps for their maintenance during adulthood [160]. Tau, coded by the *MAPT* gene, is the most abundant MAP in neurons [161]. In humans, nine isoforms of tau have been discovered, produced by alternative splicing: PNS-tau (~70 kDa, in the peripheral nervous system); fetal-tau (~37 kDa); tau A (~33 kDa); 1N3R (~40 kDa); tau-3, 2N3R (~43 kDa); 0N4R (~40 kDa); 1N4R (~43 kDa); tau-4, 2N4R (~46 kDa); and tau G (~81 kDa). The importance of tau for neurons is reflected in its involvement in axonal degeneration in several neurological disorders, particularly in Alzheimer’s disease (for a review of the role of tau in AD, see [162]). Moreover, the involvement of tau inclusions and/or structural alterations in several other pathologies has given rise to the term *tauopathies*. Among these, AD is considered a secondary or non-primary tauopathy, although given the broad spectrum of phenotypical manifestations of the combined effect of Aβ plaques and tau inclusions, this classification is still controversial [163]. Both Huntington’s disease (HD) and Parkinson disease are also related to the accumulation or abnormal processing of tau, though in none of them does tau seem to be a primary cause of the disease; details on these and other tauopathies in the context of neurodegeneration will be further discussed in Section 3.

Other members of MAPs with a key role in neurons are the Ending Binding (EB) family of proteins. EB proteins belong to the MT plus-end tracking proteins (+TIPs) and include three homologues with presumably distinctive functions: EB1, EB2/RP1, and EB3. All the EB proteins bind to the plus-end of MTs and regulate its dynamics. EB3 (encoded by *MAPRE3* gene [164]) is expressed preferentially in the nervous system [165]. Similar to EB1, EB3 binds and accumulates at the tip of MTs while they are growing and dissociates when the process of growth pauses or when it switches from growth to shrinking (MTs become smaller in size). This particular behavior of EB proteins gives them a form of “comet-like” structures. These comets can be visualized by using enhanced green fluorescent protein (EGFP)-tagged EB3 as a marker of microtubule growth in neurons [166] (see video 1) or transgenic mice expressing yellow fluorescent protein-tagged EB3 to study MTs in intact mammalian neurites [167]. EB3-comets have been observed to enter the dendritic spines of developing hippocampal neurons at the tip of growing MTs [168] and interact with a stromal interaction molecule 2 (STIM2), a neuronal-specific endoplasmic reticulum calcium sensor protein, to promote the formation and maintenance of dendritic spines [169]. EB3 is also recognized to control microtubule dynamics during axon outgrowth [170,171,172,173], and retrograde transport in axons [174]. Additionally, new knowledge shows how Cytoplasmatic Linker Protein 170 (CLIP-170), a member of the +TIPs family, coordinate microtubule dynamics in neurons. CLIP-170, in a mechanism dependent on the canonical plus-end tracking EB proteins, regulates organelle transport initiation in neurons [175].

#### 1.3.2. Post-Translational Modifications of Tubulin (PMT)

The functional properties of MTs are also modified through the addition of a chemical group on unpolymerized or polymerized tubulin subunits. In particular, PMT has been an active focus of research for neurological disorders. Among the reported PMTs, tubulin tyrosination/des-tyrosination [176], tubulin glutamylation [177,178], and tubulin acetylation [179,180] is the most studied in neurons (for recent detailed reviews, see [181,182]). In particular, tubulin acetylation and tubulin tyrosination/detyrosination have an important effect on the dynamic and mechanical properties of MTs but have also been shown to be critical for the binding of motor proteins to the MTs.

The addition of the Tyrosine (Tyr) group takes place on the dimer of α/β-tubulin and is a reversible process. The recently identified tubulin carboxypeptidase (TCP) (a complex of vasohibin-1, VASH1, with the small vasohibin binding protein, SVBP) [183] starts the cycle of tyrosination/detyrosination with the removal of Tyr in the carboxyl-terminal end of the α-tubulin (Figure 8). Then, the cycle is completed by the re-addition of Tyr by the tubulin tyrosine ligase (TTL) [184], an enzyme whose activity depends on Mg^2+^ and ATP and that only works on the soluble dimers of α/β- tubulin. In neurons, the cycle of the tyrosination/detyrosination of tubulin modulates several functions of developing and mature axons. The tyrosination of tubulin has been observed mostly in new and more dynamic MTs, which usually accumulate in the growth cones [185]. An increase of tyrosination is essential to form and maintain the motility of the axonal growth and also as a signal to activate the regenerative response of peripheral axons after an injury [186]. In addition to this evidence, α Tubulin tyrosination has been found to enhance the efficiency of the cargo-binding to MTs [175]. On the other hand, detyrosinated tubulin accumulates in mature axons [180,187], where it is thought to facilitate the anterograde axonal transport of the motor protein kinesin KIF5 towards the most distal part of the axon [188]. Moreover, other studies support the hypothesis that α tubulin detyrosination plays a role in spatially defining retrograde axonal transport initiation in the distal axon [175,186].

α-Tubulin acetylation occurs mainly on the polymers of α/β-tubulin and consists in the addition of an acetyl functional group at the Lysine 40 residue of α-tubulin (αK40) located at the amino-terminal end of α-tubulin (Figure 8) [189]. Enzymes such as the N-acetyltransferase complex (ARD1-NAT1) [190], the elongator acetyltransferase complex (ELP) [191], and the major mammalian α-tubulin N-acetyltransferase 1 (αTAT1) [192] catalyze this reaction under several conditions. On the contrary, other enzymes can remove the acetyl group previously added at α K40. The human histone deacetylase 6 (HDAC6) [193,194] and Sirtuin 2 (SIR2) [195] catalyze these reactions in neurons preferentially. α-Tubulin acetylation is abundant in MTs of mature axons, the same that exhibit abundant detyrosination [180]. This PMT promotes the binding of motor proteins, such as dynein [196] and kinesin [197,198], to the MTs. Additionally, it increases the resistance of MTs to breakage [199], a mechanical property essential to maintain the stability of the MTs in axons, particularly in those of the peripheral nervous system that can reach lengths of 1 m in humans [200,201]. Since the axonal transport is essential for neuronal development, the maintenance of mature neurons during adulthood, and the regeneration of peripheral axons α-tubulin acetylation is recognized to be an essential PMT for the homeostasis of mature neurons [179,202].

### 1.4. Cytolinker Proteins

Plakins or cytolinker proteins are a family of seven cytoskeletal cross-linker proteins able to connect the different elements of the cytoskeleton with each other and to junctional complexes. In mammals, the plakin family is composed of: bullous pemphigoid antigen 1 (BPAG1), microtubule actin crosslinking factor 1 (MACF1), plectin, desmoplakin, envoplakin, periplakin, and epiplakin. These cytoskeletal proteins are expressed in tissues that undergo mechanical stress, such as epithelial and muscle tissues, and peripheral nerves [203].

BPAG1, known as dystonin, and MACF1, also called spectroflakin, connect the cytoskeleton with proteins that have an important role in cell migration, the maintenance of cell morphology and cell–cell contact. Moreover, they are critical in the development and survival of neurons. That is why it is thought that the lack of dystonin or MACF1 is incompatible with life. Only one mutation in the *DST* gene, which is associated with a severe postnatal neurodegeneration, has been described [204] that will be further discussed in Section 3. MACF1 is related with the formation and function of neuromuscular synapses, maintaining an efficient synaptic transmission, and mutations in this gene are associated with congenital myasthenia in humans [205].

Plectin and dystonin are able to bind actin microfilaments, microtubules, and integrins, but plectin is primarily an intermediate filament (IF)-binding protein. Plectin anchors IFs to multiple cellular sites, such as mitochondria and the nuclear/ER membrane system, as well as the desmosomes and hemidesmosomes of epithelial cells and the Z-disk of skeletal muscle fibers, thus intervening in the formation of these structures and demonstrating their importance for skin and skeletal muscle integrity. Plectin is expressed in a wide variety of cell types and the differential subcellular targeting is due to the existence of various isoforms that vary just in their short N-terminal sequences. The first functions described for these proteins were the mechanical stabilization of cells and the regulation of cytoskeletal dynamics. Moreover, the existence of multiple functional domains around the protein facilitates an interaction with many other proteins. Concretely, the complex of plectin with the receptor for activated C kinase 1 (RACK1) regulates protein kinase C activity that involves plectin in intracellular signaling cascades [206].

## 2. Axonal Growth and the Growth Cone

Axonal growth is a key process for the formation of the human nervous system during embryonic development. In the case of the neurons of the peripheral nervous system (PNS), axonal growth is also important during adulthood since these neurons, especially neurons of the dorsal root ganglion (DRG), are recognized by their intrinsic ability to regenerate their axons [207]. Axonal growth requires the orchestration of several molecular and cellular mechanisms. The key organelle is the neuronal cytoskeleton, together with mitochondria and the endoplasmic reticulum that provide energy and materials to supply the new axon (e.g., synthesis of lipids and proteins). However, mitochondrial networks are essential because they contribute to adjusting ATP, Ca^2+^, and reactive oxygen species levels that are required to regulate the activity of cytoskeletal proteins during axonal growth [208,209].

During neurite outgrowth, central and peripheral axons travel variable distances and navigate through tissues in order to reach their targets. The structure responsible for this important task is the growth cone (GC). The GC is a sensitive and motile structure, whose morphology changes quickly. At the structural level, the GC is recognized by two parts: a widely scattered and flattened structure called *lamellipodia* and an extension of peaks called *filopodia* (Figure 9). Taking into account the molecular organization of the cytoskeleton within the GC, its structure can be divided into three regions: a central domain (C) rich in MTs; a peripheral domain (P) enriched with actin filaments; and a transition zone, consisting of an area at the interface of the P and C domains (Figure 10). The axonal growth occurs in three phases: first, the protrusion, then obstruction, and, finally, the consolidation. The protrusion is the extension of the *lamellipodia* and the *filopodia*; the obstruction is the swelling of the *lamellipodia* and the *filopodio* while the growth cone swallows them. The consolidation is the active narrowing of the growth cone when becoming the new axon (see Appendix A). These three phases occur continuously while the GC advances.

In general, the sensitive and explorative ability of the GC relies on the continuous reorganization of actin and MTs that allow the GC to extend, to retract, and to turn to the direction of growth. Thus, the dynamics of these cytoskeletal components allow the growth cone to explore its environment to form the new axon properly. The behavior of the GC is a dynamic event that occurs through the coordination of actin and microtubule cytoskeletons and also involves regulatory mechanisms that remain poorly understood.

### 2.1. Actin Dynamics in the Growth Cone (GC)

Actin dynamics are important to the GC for two reasons: firstly, to provide the necessary forces for the protrusion of the GC, through which the *filopodio* and the *lamellipodia* can explore the environment. Moreover, secondly, to generate the traction forces in the P domain of the GC that take the GC forwards and directs the GC’s turning (see Appendix A).

Within the *filopodia*, the polymerization and recycling of actin continuously reorganize actin filaments. Recycling occurs through a process called *retrograde F-actin flow*, which is the retrograde translocation of filaments at the barbed end that are later recycled through the disassembly of the actin filament by the action of cofilin and profilin (Figure 10). The GC advances when the polymerization speed of actin at the leading end exceeds the retrograde F-actin flow (v_average_~ 3-6 µm/min) (Figure 10) [129,210]. The retrograde F actin flow is directed by the motor protein myosin II, which generates traction forces enabling the GC to be rowed forward [211]. Thus, GC dynamics is modulated by the same proteins that regulate the polymerization of actin and the retrograde F-actin flow.

Among the proteins that control the polymerization of actin in the leading end of the *filopodia,* we find actin nucleating proteins: the ARP2/3 complex and formins. The ARP2/3 complex is abundant within the C domain and T zone of the GC and is important for the formation of *lamellipodia* and *filopodia* [212]. Formin promotes nucleation for long filaments of actin, which are necessary for the formation of the *filopodium*, and probably also promotes the integrity of the actin bundles within *filopodia* [213]. However, the role of formins in the growth cone seems to go beyond these functions. The Dishevelled-associated activator of morphogenesis (DAAM), a protein member of formins, binds EB1 (a +TIP protein) to coordinate actin and microtubule communication within the axonal GC [214]. It is thought that a formin/+TIP-dependent mechanism is crucial for leading actin and MTs coordination in growing axons [214]. In contrast to the polymerization of actin, the retrograde F actin flow and the disassembly of the actin filaments in GCs are regulated by the RhoA GTPase signaling. The Rho-associated protein kinase (ROCK) is one of the effector proteins of the RhoA GTPase signaling. ROCK regulates the activity of other protein kinases with specific functions downstream RhoA GTPase: i) MLCK (Myosin Light Chain Kinase): phosphorylation of MLCK activates myosin II and promotes its association with actin filaments managing the retrograde F-actin flow; ii) LIMK (Protein kinase domain LIM): when LIMK is activated, it phosphorylates and inactivates ADF/cofilin, which promotes the destabilization and break of the actin filaments; iii) ERM (Ezrin, Radixin, and Moesin) complex: the proteins of the ERM complex are actin-binding proteins that when active (phosphorylated) tie actin filaments to the cell membrane. Specific adhesion receptors of the GC, such as L1CAM, regulate the activity of the ERM complex. ERM proteins are important for the renovation of actin filaments in response to the activity of neurotrophic factors, such as neurotrophins (factors that favor the survival of neurons) and possibly the redistribution of attachment receptors of the GC to the substrate [215].

### 2.2. Actin Dynamics in Axons

Actin dynamics into the growth cone have been a focus of active research, since it was the only neuronal structure where actin dynamics could be observed. New advanced microscopy techniques have revealed novel surprising actin structures (for a recent review of actin within axons, see [216]). Actin rings were the first major actin structure in axons. Now, two more actin structures have been recognized: waves (actin waves), and trails of actin (actin trails) (Figure 11, center). Actin rings are well known to form a cylindrical and periodic structure under the axonal plasma membrane and dendrites [217]. Actin filaments stabilized by an encapsulating protein (adducin) form each ring and are separated between them through spectrin multimers, an actin-binding protein [218] (Figure 11, left). This organization of actin makes the rings stable and give shape to the axon membrane skeleton. Actin waves are structures similar to the GCs, with the difference that they emerge at the base of the neurites, migrating slowly to the tip and protruding on the plasma membrane (Figure 11, center) [130,131,219]. Actin waves exhibit a slow movement (2–3 µm/min) and aperiodic motion, with 1–2 waves appearing every hour. Within an actin wave, actin and MT filaments are present [131,132], and G-actin is added at the leading end of the filament and is disassembled at the base, similar to what is observed in the GC. Furthermore, doublecortin (DCX), a stabilizing protein of the cytoskeleton that binds both actin and MTs, is responsible for the communication of actin and MTs within the actin wave [220]. Regarding the functional properties of actin waves, there seem to be transport mechanisms that bring actin and ABPs to the GCs. Additionally, it has been proposed that they can promote anterograde microtubule and kinesin-based transport in the neurite outgrowth during the multi-polar state before an axon is specified [221]. Finally, actin trails are the most recently described structure inside the axon (Figure 11, right) [148], and the phenomenon occurs at certain points where actin filaments undergo continuous polymerization and depolymerization cycles. The separation between trails is about 2–3 µm [148], and their function consists in the formation of a nest for the fast assembly of actin filaments. In this nest, the long filaments accelerate with the bidirectional movement along the axonal axis [148]. The assembly of these filaments possibly starts in stationary endosomes within axons and seems to depend on the actin-nucleating protein *formin*, but not on the ARP2/3 complex [148].

### 2.3. Microtubules in the Growth Cone (GC)

MTs arriving into the GC form several arrangements (spread, bonded, and bundled) and together with actin filaments determine the morphology of the GC. The main function of MTs in the GC is related to the decision of the trajectory of the GC. When the GC turns, the MTs are grouped in the central domain of the growth cone and then scatter and penetrate the transition zone and invade the P domain preferably in the area where the growth cone is going to turn [222] (Figure 12). In order for MTs to enter the P domain, actin must reorganize or depolymerize in certain areas of the GC. When the GC moves, the retrograde F-actin flow produces an attractive force that pushes them towards the C domain of the GC [223,224] (Figure 12B). Actin-based (e.g., Myosin II) and microtubule-based motor proteins (e.g., dynein and kinesin) generate ATP-dependent forces governing the dynamics of actin filaments and MTs as well as the communication between them within the GCs. It has been shown that MTs can use these forces to beat the retrograde F-actin flow and thus successfully invade the P domain (for a review see [225]) (Figure 12C,D). Two intracellular signaling pathways have emerged as candidates for the regulation of the dynamics of MTs in axons and within the GC: Rho GTPase, a family of proteins that also participate in the regulation of the actin cytoskeleton [226,227] (for a review see [228]); and the glycogen synthase kinase-3 (GSK-3), which is a protein kinase in the WNT signaling pathway (for a review see [229]).

As presented before, the cytoskeleton, as well as its dynamics, is essential for the development and the function of the nervous system but is also particularly important for its maintenance during adulthood. In addition to this, there is a shred of increasing evidence that shows how defects of the neuronal cytoskeleton can lie at the core of neurological diseases. Therefore, in the next section, we review relevant evidence that links defects of cytoskeletal proteins and neurological diseases.

## 3. Cytoskeletal Abnormalities in Neurological Diseases

The function of cytoskeletal proteins is essential to develop and maintain the function of the nervous system. Since their formation, human neurons and, in particular, the peripheral nervous system, are continually exposed to physical, chemical, and biological damaging agents that can produce abnormalities in cytoskeletal proteins. Several inherited neurodegenerative diseases involve the destabilization of cytoskeletal proteins and their dynamics. These diseases are associated with mutations in cytoskeletal genes (Table 2) and can be considered as a primary cause of the disease. Additionally, mutations in cytoskeletal non-related genes promote the cytoskeleton destabilization, contributing to the pathophysiology of the disease (Table 3). In any case, both can affect the structure or dynamics of the three cytoskeletal filaments in central or peripheral neurons by different mechanisms.

### 3.1. Mutations in Cytoskeletal Genes Associated with Neurodegenerative Diseases

In the present section, we will review those pathological processes in which cytoskeletal changes are primary causes of the disease, mainly due to the mutation in genes coding for specific cytoskeletal proteins.

Mutations on actin and/or its regulatory proteins (such as ABPs) can easily change the organization of actin cytoskeleton in neurons. Actin filament destabilization can produce several pathological effects in brain neurons: impairment of neurite outgrowth during neuronal development (a function of the growth cone), the pathological formation of axonal inclusions—“action rods”—and abnormal formation of dendritic spines and synapses. Several of these abnormalities are a consequence of the dysfunction of the actin cytoskeleton regulatory proteins that depend on the redox balance and ATP levels to function properly. It is worth mentioning that most of the evidence supporting the role of these cytoskeletal abnormalities in human disease came from the study of neurons from the central nervous system, but this kind of abnormality can also be found in peripheral neurons. The identification of profilin 1 (PFN1) as a causative of familial amyotrophic lateral sclerosis (fALS) [231,269,270] or sporadic ALS [231,232,233] confirmed that the cytoskeletal pathway alterations are contributing to ALS pathogenesis. ALS causes the progressive death of motor neurons in the brain and spinal cord. Mutations in PFN1 decrease bound actin and inhibit axon outgrowth in motor neurons [231]; other mutations cause detergent-resistant protein aggregates, which is a hallmark of ALS [270] and some mutations abolish a proven phosphorylation site in profilin 1 preventing the interaction with its binding partners [269]. Spinocerebellar ataxia type 5 (SCA5) is an inherited neurodegenerative disease associated with mutations in the *SPTBN2* gene that encodes spectrin [271]. The mutation L253P localizes in the actin-binding domain of spectrin, causing high-affinity actin-binding [272]. Then spectrin–actin dynamics decrease, and during axon arbor growth expansion reduces the stabilization of distal dendrites required to correct Purkinje cell dendritic arborization [234]. Mutations in the inverted formin-2 (INF2) gene were identified in patients with Charcot-Marie-Tooth (CMT) disease combined with focal segmental glomerulosclerosis (FSGS) [239,240,241,242]. INF2 mutations promote actin cytoskeleton dysfunction, disturbing the polarization of Schwann cells. These mutations cause abnormalities in the myelination, and the axonal loss observed in these patients is probably secondary to Schwann cell pathology [241].

Mutations or abnormal regulatory mechanisms that change the structure and dynamics of cytoskeletal proteins tubulin or MAPs can induce the disruption of MTs and therefore produce the blocking of axonal transport. Neurons, especially those belonging to the peripheral nervous system, require axonal transport to maintain their long axons functional. Therefore, abnormalities related to MTs are associated with the axonal degeneration underlying the most typical neurodegenerative diseases that involve motor and sensory neurons. The tubulinopathies or complex cortical malformations are caused by mutations in tubulin isotypes and are responsible for neurodevelopmental disorders [273]. Nevertheless, mutations in tubulins associated with neurodegenerative diseases have also been described: a novel mutation in TUBB2A causes progressive spastic ataxia and sensory-motor peripheral neuropathy [235]. This mutation impairs the affinity of MTs to neuron-specific kinesin KIF1A, unlike other mutations in this gene, and in other tubulins, which impair the correct formation of the α/β tubulin heterodimer to form microtubule polymers. Other causes of a previously unreported sensorineural disease related to tubulins are the mutations in TUBB4B. Changes in the Arg391 residue, which contribute to a correct binding with α-tubulin, cause a severe neurodegenerative condition of the retina known as Leber congenital amaurosis (LCA) with cochlear cell loss [236]. Mutations in the tubulin alpha-1A gene (*TUBA1A*) are the most common cause of the cases of tubulinophaty, and mutations in this gene cause lissencephaly as a result of the lack of cytoskeletal integrity [237]. It is nonetheless noteworthy that some reported cases, in which TUBA1A-mutation associated lissencephaly shows comorbidity with Hirschsprung disease and the syndrome of inappropriate antidiuretic hormone secretion (SIADH) [274], highlight the difficulties in diagnosis associated with a broad spectrum of genetic and phenotypic variations, all of them closely related to the cytoskeletal function in neurons.

Primary tauopathies, in which abnormal-modified tau intracellular inclusions are found in the autopsy of patients, include major forms of Frontotemporal Lobar Degeneration (FTLD) neuropathology, in which the *MAPT* gene located in chromosome 17 is affected (with more than 40 different pathogenic mutations) [275]; Progressive Supranuclear Palsy (PSP which is largely sporadic. It is noteworthy that familial and sporadic cases of tauopathies share many clinical phenotypes: for instance, Frontotemporal dementia and parkinsonism linked to chromosome 17 (FTDP-17), a familial form of FTLD initially associated with *MAPT* mutations [275], was later shown to affect also patients with mutations in the nearby gene progranulin, although they are not clearly distinguishable from FTLD-tau patients [275]. Following this line of evidence, Forrest and collaborators suggested to retire the term FTDP-17 in order to use the classification of FTLD-tau subtypes in order to differentiate familial and sporadic forms [275], which highlights the complexity of the cellular consequences directly or indirectly related to the participation of the tau protein in cytoskeleton homeostasis.

Last, but no less important, abnormalities related with intermediate filaments commonly produce the reduction of axonal caliber with a consequent reduction of the caudal nerves. Mutations in the NEFL gene encoding neurofilaments are responsible for autosomal CMT with different phenotypes: CMT1 (CMT1F), CMT2 (CMT2E), dominant intermediate CMT, and autosomal recessive CMT (for a review see [238]). A large number of NEFL mutations have been described, and the pathogenic mechanisms among them are different. Changes in NFL can affect the axonal transport of neurofilaments and mitochondria, can disrupt their ability to self-assemble into a filamentous network, which leads to the aggregation of NFL protein, and impair motor neuron viability in vitro [276,277,278]. Then, disruption of NF network with the aggregation of NFL is a common triggering event of motor neuron degeneration that contributes to the loss of large myelinated neurons of the sural nerve, a uniform feature among CMT patients.

Finally, mutations in plakin genes have also been related with neurological diseases. *DST* gene mutations result in a hereditary sensory autonomic neuropathy type VI (HSANVI) [204,243,244]. Patients with less dystonin experiment autonomic dysfunction, including cardiovascular dysregulation, altered sweating, and gastrointestinal dysmotility. Studies of nerve conduction confirmed a severe neuropathy in these patients, and skin biopsies showed a loss of sensory and autonomic nerve fibers [244]. Mutations in the plectin gene are responsible for human diseases denominated *plectinopathies* [247]. Among these, the most common is epidermolysis bullosa simplex, with muscular dystrophy and neuropathy-like symptoms due to defects in the peripheral myelin ultrastructure [246,279].

### 3.2. Neuronal Cytoskeleton Abnormalities Generate Neurodegeneration

Diverse neurodegenerative diseases exhibit severe alterations in the neuronal cytoskeleton, which affect disease progression but are not directly due to mutations in proteins related to actin, microtubule, or neurofilament proteins. Thus, in all pathologies reviewed in the present section, cytoskeletal damage can be considered as a secondary cause in the physiopathological mechanism of the disease.

A good example of this is that of frataxin, the protein responsible for Friedreich’s ataxia (FRDA). Frataxin deficiency causes cellular redox disequilibrium, probably as a result of the accumulation of reactive oxygen species generated by iron overload and one of the consequences of the resulting oxidative stress is the damage of neuronal cytoskeleton. For example, the respreading of vimentin fibrils is decreased in fibroblasts from FRDA patients and is associated with their lower growth potential in culture [251]. The fibroblasts from patients with FRDA, under oxidative stress, exhibit an increase in MT and actin glutathionylation that impairs the organization of microfilament [248] or MT dynamics [249], oxidized glutathione (GSSG) being a mediator of the damage. This abnormal microtubular polymerization associated with an increment of tyrosinated tubulin has been described in the spinal cord of FRDA patients, in addition to an irregular distribution of phosphorylated NF-H (SMI 34) [250]. All these results suggest a negative effect in the cytoskeletal network that impairs axonal transport, as reported in cultured neurons of YG8R DRG [280] and *Drosophila* models [281].

Other post-translational modifications, such as tubulin acetylation, have been widely related to CMT. Mutations in GDAP1 cause axonal recessive (AR-CMT2), axonal dominant (CMT2K), and demyelinating recessive (CMT4A) forms of CMT. Motor and sensory neurons of Gdap1 knockout mice showed significantly decreased acetylation in α-tubulin [252]. Similar post-translational modifications are observed in symptomatic neurons of transgenic mice expressing a CMT-associated mutant form of HSPB1 (CMT2F) [253].

Alzheimer’s disease (AD) is a multifactorial disease with both familial (FAD) and sporadic (SAD) forms. As mentioned above, AD is included in a group of neurodegenerative diseases denominated tauopathies, which are characterized by an altered metabolism and the deposition of the neuronal microtubule-associated protein tau. Then, one of the pathological hallmarks of AD is the presence of neurofibrillary tangles (NTFs), which are composed of a phosphorylated form of tau [255]. The hyperphosphorylation of tau proteins results in microtubule destabilization and cytoskeleton abnormalities. Apart from tau phosphorylation, other post-translational modifications of tubulin are altered in AD [258]. Interestingly, it has been shown that a reciprocal relationship between O-GlcNAcylation and tau phosphorylation regulates tau aggregation in the AD brain (for a more detailed review, see [282]). Moreover, actin aggregates in the form of cofilin–actin rods have been identified in the postmortem brains of patients with AD [283]. The abnormal regulation of cofilin promotes persistent rods, which directly block transport and represent a possible mechanism to explain the synapse loss and cognitive decline occurring in AD progression [254,256,257].

Huntington’s disease (HD) is a devastating autosomal dominant neurodegenerative disorder that courses with a combination of motor, cognitive, and psychiatric symptoms, atrophy of the basal ganglia, and the cerebral cortex. HD is caused by an increase in the number of copies of the glutamine encoding CAG repeats in the exon 1 of the *HTT* gene, encoding the protein huntingtin (HTT) [284]. HD is considered as a tauopathy, since the total amount of tau is increased in the cortex of HD patients and correlate with mutant huntingtin levels [265]. The splicing factor SRSF6 that affects the alternative splicing of exon 10 from the tau gene (*MAPT*), is altered in the brain of HD, producing an imbalance in tau mRNA isoforms. Besides *MAPT*, other microtubule-associated proteins (MAP2), mainly in dendrites, are coded into a target of SRSF6. The different MAP2 isoforms could explain the dendritic alteration of striatal medium spiny neurons in patients. HTT plays an important role in neuronal transport. The interaction of HTT with dynein and HAP1, proteins that regulate the transport of organelles, have been reported, which, when the complex is disturbed, lead to dysfunction of the axonal transport (for a review see [285,286]).

Another polyQ neurodegenerative disease closely related to HD is Spinocerebellar Ataxia type 3 (SCA3), also known as Machado–Joseph disease. As in HD, an expansion on the CAG triplet producing long polyglutamine stretches is at the base of the molecular alterations underlying SCA3, but in this case, the expansions affect the *ATXN3* gene. Besides, a common treatment of both diseases is a disruption of the actin cytoskeleton that affects dendrite arborization, in which the GTPase Rac.V12 and the G protein formin 3 have been shown to play a critical role in *Drosophila* models of both pathologies [266].

Parkinson’s disease (PD) is characterized by the loss of dopaminergic neurons of substantia nigra pars compacta (SNc) and the presence of cytoplasmic inclusions named Lewy bodies (LB) (for a review see [260]). LB contain tubulin and neurofilaments, suggesting abnormal neuronal cytoskeleton function. Moreover, diverse studies have shown that the PD-associated proteins, like parkin, PTEN induced putative kinase 1 (PINK1), α-synuclein or Leucine-rich repeat kinase 2 (LRRK2) may modify microtubule stability. Parkin is an E3 ubiquitin ligase that enhances tubulin ubiquitination [262], stabilizes MT through strong binding [264], and protects dopaminergic neurons against depolymerization via MAP Kinase activity [261]. Both PINK1 and parkin are related to mitochondrial motility [263], and mutations in these proteins promote changes in mitochondrial distribution, transport, and dynamics. LRRK2 is a complex Kinase/GTPase that interacts physically and functionally with the cytoskeleton, and it is related with neurite outgrowth, axonal transport, and synaptic vesicle trafficking (for a review see [259]).

It is interesting to note that alterations in neuronal cytoskeleton physiology can also be used to monitor disease progression since they can even be detected in pre-clinical stages of specific neurodegenerative diseases. This is the case with the MM1 and VV2 subtypes of sporadic Creutzfeldt-Jakob disease (CJD), where cytoskeleton assembly dysregulation mediated by alterations in cofilin-1 and its upstream regulators LIMK1, SSH1, Rock2, and APP has been described in both mice and humans [268]. Importantly, the finding of mechanistic relationships between cofilin-1 phosphorylation, its interaction with the prion protein PrP^C^, and alterations in activated microglia, as well as the accumulation of a dense form of F-actin, were found in frontal cortex and cerebellum samples from individuals at the pre-clinical stage, highlighting the potential for early diagnosis held by the analysis of cytoskeleton alterations in neurodegeneration. Supporting this notion, the finding of higher levels of tau and NF-L proteins in both plasma and cerebrospinal fluid (CSF) in CJD and AD patients suggests the possibility of using the detection of plasma cytoskeleton proteins as a tool to detect neurodegeneration, and to contribute and support early diagnosis and the identification of disease subtypes [267].

Non-cytoskeletal proteins that are very similar at the functional level are the motor proteins, which perform intracellular transport processes, and their impaired function constitutes another critical factor in the development of neurodegeneration. For instance, mutations in dynein motors have been described at the basis of many human motor neuropathies, including spinal muscular atrophy (SMA) or CMT, but also malformations of cortical development like lissencephaly. Mutations in the protein LIS-1 (*PAFAH1B*), part of the dynactin complex which binds to and regulates dynein function, produces lissencephaly, a condition of severe brain abnormalities in which neuronal migration is drastically affected. Different LIS-1 mutations give rise to a spectrum of phenotypes with graded severity [287], being the duplications that included LIS-1 and adjacent genes the strongest. *MACF1* gene mutations also cause a rare lissencephaly with complex brainstem malformation [245]. The severity of the phenotype depends on the spectrum of mutations: patients with mutations located in the spectrin repeat domain present subtle brainstem dysplasia as compared to patients with variants located in the GAR domain or nearby, which affect the microtubule binding site. *MACF1* has been involved in the pathogenesis of Parkinson disease, since it has been observed that PD patients have lower *MACF1* mRNA levels than controls [288], leading to dysregulation of the cytoskeleton in dopaminergic neurons. Very recently, a new hypothesis has been proposed that involves *MACF1* with Alzheimer’s disease. The hypothesis proposes that the extracellular amyloid-β (Aβ), a hallmark of AD, induces the phosphorylation of MACF1 and leads to its association to the microtubule, altering synaptic structure and activity [289].

One of the most intriguing relationships between cytoskeleton structure, intracellular trafficking, and human neurodegenerative disease can be found in Lowe syndrome (LS), a rare genetic condition which leads to progressive oculocerebrorenal defects. The gene responsible for LS is *OCRL1* (Oculo-Cerebro-renal syndrome of Lowe 1) [290], encoding a phosphoinositide-modifying enzyme critical in the cellular process of cytokinesis. Interestingly, Ocrl1 inactivation triggers abnormal F-actin dynamics in interphase cells, affecting endocytic recycling. The intracellular roles of Ocrl1 are multiple since it interacts with a wide array of proteins, thus giving rise to different diseases apart from Lowe syndrome. However, from a neurodegeneration-based point of view, it seems plausible that most of the neuronal defects observed in LS patients could be related mainly with the abnormal endocytic recycling affecting cell polarity and the turnover of membrane receptors in neurons. Nonetheless, more in vivo evidence is required, at the current moment, to unveil the specific mechanisms giving rise to the complex disease phenotype [291]. In any case, this latter example corroborates that the role of cytoskeleton function and dynamics must be necessarily taken into account when studying neurodegenerative diseases, even in those cases in which the genes involved seem to be unrelated.

In conclusion, even though cytoskeletal changes are considered secondary features and not the primary cause of disease in all the aforementioned pathologies, it is evident that cytoskeletal function contributes largely to the pathological phenotype of these diseases.

## 4. Conclusions and Perspectives

It is clear from this review that the cytoskeleton is an extremely complex organelle. This complexity is easy to understand. Neurons are rather peculiar cells with extremely long processes that play roles of paramount importance. Axons are very dynamic structures, a property which is reflected in the fact that they can even regenerate in some circumstances. This capacity is largely dependent on cytoskeleton structure and dynamics, which grant the versatility of the growth cone responding to its environment. Since cytoskeletal proteins and regulators also perform other critical roles in neuronal function (axonal transport, neurotransmitter secretion, etc.), they have become increasingly involved in devastating diseases, most of them without actual treatment. Moreover, the wide spectrum of cellular symptoms, which expand from subtle alterations in cytoskeleton-related proteins, gives rise to very complex familial or sporadic syndromes that share biochemical features and are thus complicated to identify, making it difficult to predict the progression of the consequent neurological deterioration. This fact is clearly exemplified in the biochemical complexity, genetic heterogeneity and phenotypic variation underlying the multiple diseases classified as tauopathies, all of which share the common denominator of the altered structure or function of the microtubule-associated protein tau. Thus, it can be inferred from the examples provided herein that cytoskeletal changes, especially those due to hereditary mutations, are the primary cause of many neurodegenerative diseases. However, the fact that many of these diseases are chronic and have a long pathogenic procedure makes it sometimes difficult to dissect if the cytoskeletal changes described are a cause or consequence of alteration in the physiopathological machinery.

In summary, although we are currently aware that actin, MTs, and neurofilaments work in a coordinated fashion, which allows the maintenance of their function through life, we are just starting to arm the puzzle of the molecular mechanisms that govern the fine coordination between these elements inside axons. Furthermore, emerging evidence suggests that additional key enzymes could have an important role in the molecular mechanisms underlying neurological disorders [292,293], which are sometimes involved in molecular signaling pathways unrelated to cytoskeleton proteins and function. Therefore, it would take some substantial effort to understand how the coordination of the three cytoskeletal elements is affected in neurological diseases, as well as the specific role that cytoskeletal abnormalities play in the final manifestation of the complex neurodegenerative phenotypes. A deep understanding of these cytoskeletal regulatory elements is a key factor to develop new therapeutic strategies for an important number of neurodegenerative diseases that currently lack treatment.

## Figures and Tables

**Figure 1 cells-09-00358-f001:**
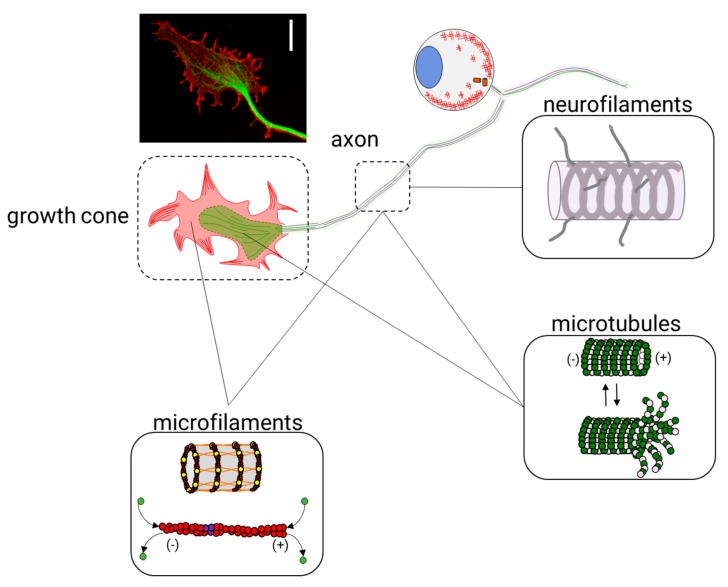
Main elements of the neuronal cytoskeleton. The neuronal cytoskeleton is mainly composed of neurofilaments, actin filaments, and microtubules. In neurons, actin filaments and microtubules are very dynamic in response to physiological needs, for example, during the embryonic development of the nervous system and the axonal regeneration of peripheral nerves after damage. Here, neurons need to grow a new axon and direct it to their right target. For this particular task, neurons use the growth cone, a motile structure rich in actin and microtubules as well as other cytoskeletal components. Scale bar 10 μm.

**Figure 2 cells-09-00358-f002:**
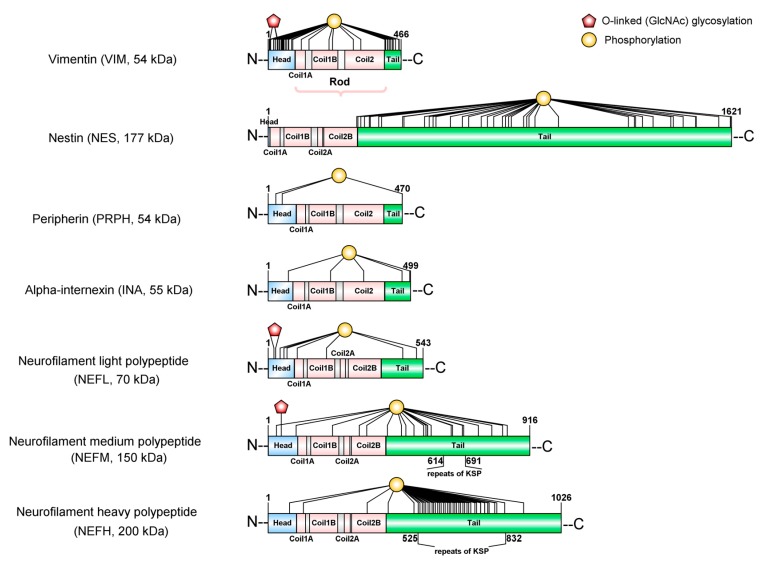
Schematic representation of the domains and posttranslational modifications of intermediate filaments. Vimentin and nestin are the subunits of neurofilaments (NFs) in the developing nervous system. Neurofilament light (NF-L), neurofilament middle (NF-M), neurofilament heavy (NF-H), α-internexin, and peripherin are subunits of NFs in the mature nervous system. All subunits share a conserved molecular structure composed by (i) an alpha-helical coiled-coil rod domain; (ii) a variable globular head at the N-terminal; (iii) and C-terminal tail domains. The chemical properties neurofilament subunits are modulated mostly by two posttranslational modifications: phosphorylation and O-GlcNAcylation. The tail domains of NF-M and NF-H are subject to extensive phosphorylation on Lysine-Serine-Proline repeats (KSP). NF proteins (NF-L, NF-M, NF-H) and vimentin are glycosylated by O-linked β-*N*-acetylglucosamine (O-GlcNAc) at the hydroxyl groups of serine (Ser) or threonine (Thr) residues of their head domains. Phosphorylation and O-GlcNAcylation residues are shown.

**Figure 3 cells-09-00358-f003:**
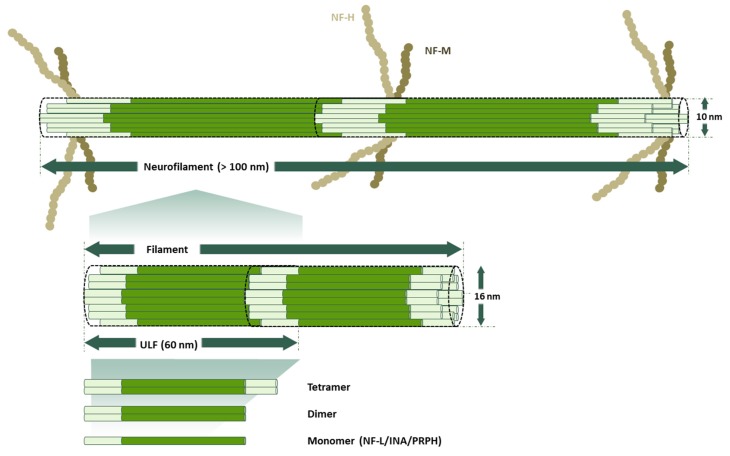
Self-assembly of neurofilaments. Monomers of NF-L, α-internexin (INA), or peripherin (PRPH) follow sequential steps to assemble into neurofilaments. First, monomers form head-to-tail coiled-coil dimers. Next, these dimers are aligned with opposite orientation to form a stable tetrameric complex without polarity. These tetramers then link end-to-end to form the unit-length filament (ULF), which expands 60 nm in length and 16 nm in width. Combinations of ULFs give rise to the filament. Finally, these filaments are helically twisted to build neurofilaments with a 10 nm diameter and more than 100 µm in length [29]. Mature neurofilaments are modified by the continuous turnover of NFs, the lateral NF subunit exchange, and the incorporation of new, shorter filaments by the interaction with any other of the neuron-specific Intermediate filaments (IFs), such as NF-M or NF-H.

**Figure 4 cells-09-00358-f004:**
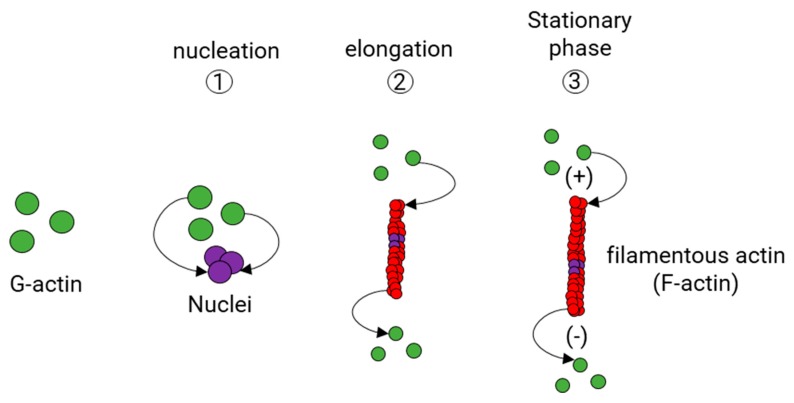
Polymerization of actin filaments. First, actin monomers linked to adenosine triphosphate (ATP) (G-actin, green) form an aggregate (violet) that grows exponentially by the addition of monomers to both ends of the filament (elongation phase). In the end, actin filaments reach a stationary state with G-actin, and actin filaments have a double-helical conformation.

**Figure 5 cells-09-00358-f005:**
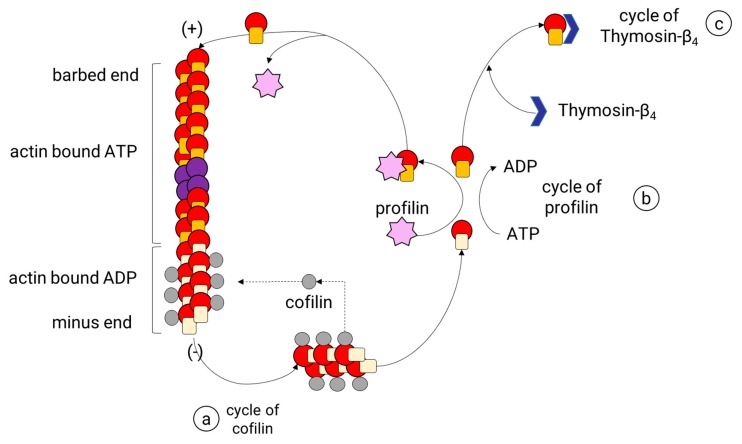
Actin-binding proteins that regulate the assembly and disassembly of actin filaments. In the cofilin cycle (a), cofilin binds preferentially to actin filaments that have actin coupled to adenosine diphosphate (ADP) (the minus end) and produce the fragmentation of actin filaments, which increases the depolymerization of the filament and leaves more minus ends exposed. In the profilin cycle (b), profilin binds to the G-actin coupled to ADP G and catalyzes the exchange of ATP to ADP. The ATP-G-actin-profilin complex can be linked to the plus end of the filament or it can also be dissociated. In the Thymosin-β4 (Tβ4) cycle (c), Tβ4 binds to ATP-G-actin subunits, inhibiting the addition of new subunits to either the plus or minus ends of the actin filament.

**Figure 6 cells-09-00358-f006:**
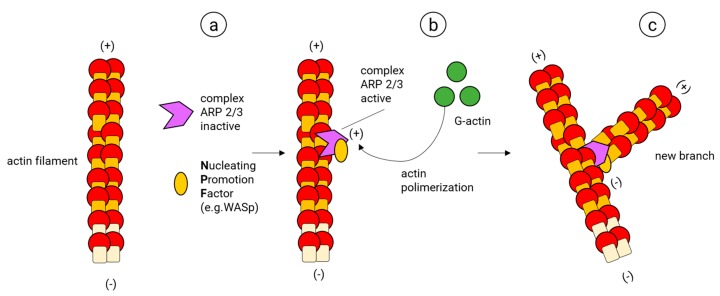
Beginning of the branches of the actin filaments by the ARP2/3 complex. The ARP2/3 complex binds to a preformed actin filament in the presence of an activating protein near to the plus end of the actin filament and then it initiates the formation of the branches.

**Figure 7 cells-09-00358-f007:**
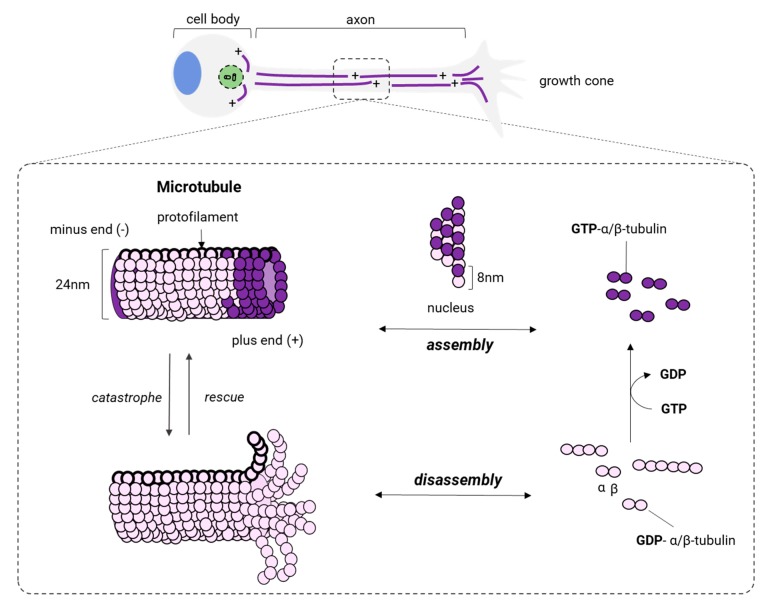
Microtubules assembly. In neurons, microtubules are assembled from the centrosome, which is located in the neuronal soma, and then they are released to be transported towards a mature axon or the growth cone of a growing neurite. On the one hand, microtubules in an axon have different sizes, but they have the same polarity; on the other hand, the microtubules in dendrites have a mixed polarity. While microtubules are transported through an axon, they suffer several cycles of growth and shortening in a process known as microtubule assembly dynamics. This process is summarized in the center of the image. The dimers of α/β-tubulin bound to GTP of the protofilament provoke the growth of the microtubule, whereas the dimers of α/β-tubulin bound to GDP of the protofilament favors disassembly. Microtubule binding proteins regulate the exchange rate between growth (rescue) and shortening (catastrophe).

**Figure 8 cells-09-00358-f008:**
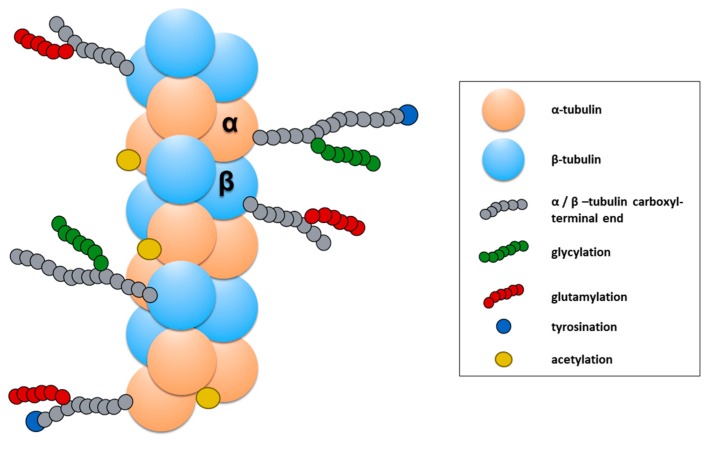
Post-translational modifications of tubulin (PMT)**.** The image shows a segment of a microtubule formed by α/β-tubulin and its posttranslational modifications. The carboxyl terminals of α/β subunits are represented as a gray chain of amino acids. Both α and β-tubulin can be modified at these carboxyl ends by polyglycylation (green chains) or polyglutamylation (red chains). De- or re-tyrosination (blue circle) are specific modifications also at the carboxyl-terminal of tubulin. In contrast, acetylation (yellow circle) takes place in the amino-terminal domain of α-tubulin, especifically at the Lys40 residue.

**Figure 9 cells-09-00358-f009:**
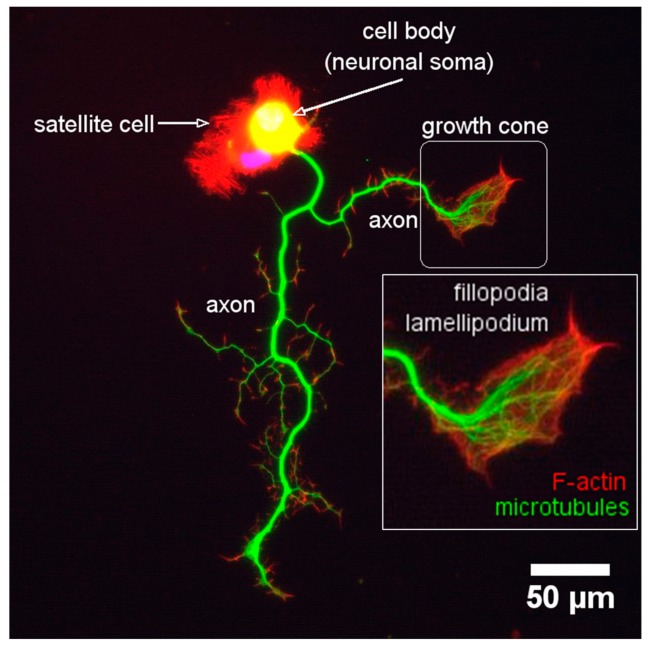
Macrostructure of the growth cone. Image shows an adult murine sensory neuron (left) in culture for 24 h. The neuron is stained with Rhodamine, Phalloidin anti-tubulin-βIII, and DAPI to identify filamentous actin (red), neuronal microtubules (green), and the nucleus, respectively. The growth cone (right-corner) is localized at the most distal part of growing neurites and axon. The growth cone is a motile and highly sensitive structure, and two parts characterize its morphology: a broad scattered and flattened structure called lamellipodium, and an extension of sharp-edged peaks called filopodium. Scale bar 50 μm.

**Figure 10 cells-09-00358-f010:**
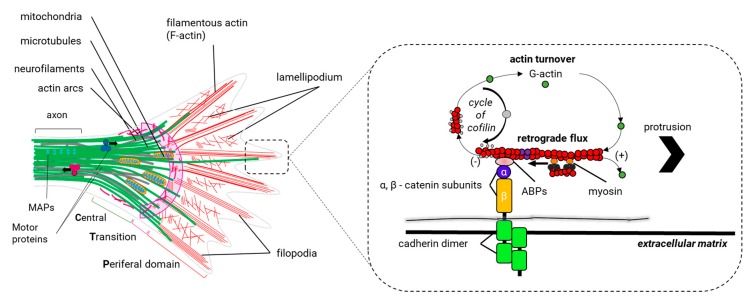
Molecular structure of the cytoskeleton in the growth cone. The growth cone can vary its shape and size very fast, but usually, three domains are distinguished: the central (C) and peripheral (P) domains, and an intermediate zone called the transition zone (T) (left). The growth cone is an expanded and motile structure where several regions can be distinguished. In the axonal axis, microtubules (MTs) (green lines) are organized in parallel bundles by the microtubule-binding proteins (MAPs), such as tau (cyan spheres), whereas in the C domain, MTs are dispersed and are extended individually through the T zone until they reach the P domain. Here, MTs are aligned with the bundles of actin filaments (F-actin, red lines) in the filopodium. Here, actin filaments are organized in parallel beams that are extended back through the P domain to end up in the T zone, where they are cut in short filaments by a still unknown mechanism. F-actin in the lamellipodium is organized in branched networks. Actin arcs (fuchsia lines), composed by anti-parallel beams of F-actin and the motor protein myosin II (not shown in the graph) are located in the T zone and at the periphery of the C domain. Actin arcs produce the compressive forces in the C domain that bend the scattered MTs and facilitate their grouping in the wrist of the growth cone. Motor proteins supply the growth cone with the materials and organelles necessary for the formation of the new axonal segment (anterograde transport) and they transport vesicles involved in the endocytic pathways (retrograde transport). In the filopodia (right), the balance between polymerization and the retrograde flow of F-actin controls the protrusion of the growth cone. F-actin is disassembled in the T zone and is polymerized in the filopodia of the growth cone. The protrusion phase occurs when the polymerization rate of F-actin exceeds that of the retrograde flow. Once the filopodium recognizes an adhesive substrate in the extracellular matrix, it binds to it using adhesion proteins such as cadherins, which in turn interact with heterodimeric transmembrane receptors of α/β—integrin in the growth cone. These receptors interact with some actin-binding proteins (e.g., vinculin, talin, paxillin) to communicate F-actin with the extracellular matrix. Src protein kinase and the focal adhesion kinase (FAK) bind to actin-binding proteins (ABPs) to activate the adhesion of other additional proteins (not shown in this figure). It is believed that the union of the ABPs restricts the retrograde flow and strengthens the polymerization of actin to generate the protrusion of the membrane.

**Figure 11 cells-09-00358-f011:**
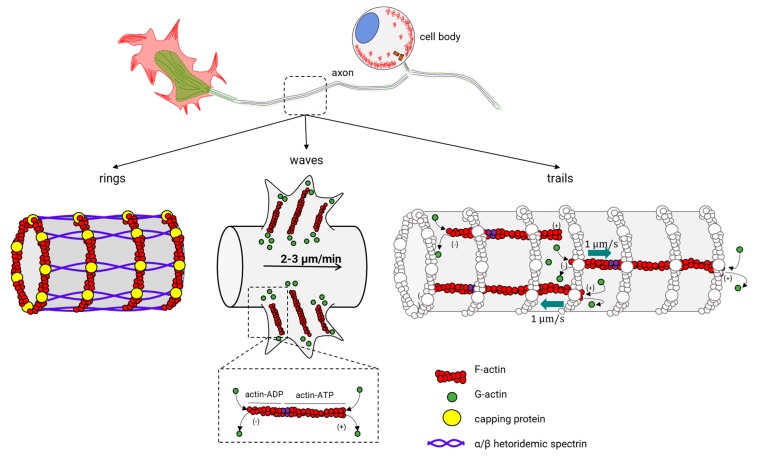
Actin structures in the axon. The figure shows three assemblies of actin in the axon, as described in the literature: actin rings (left), actin waves (center), and actin trails (right). The black arrow in the center indicates the anterograde movement of the actin waves. The green arrows on the right indicate the direction of the polymerization of the actin filaments.

**Figure 12 cells-09-00358-f012:**
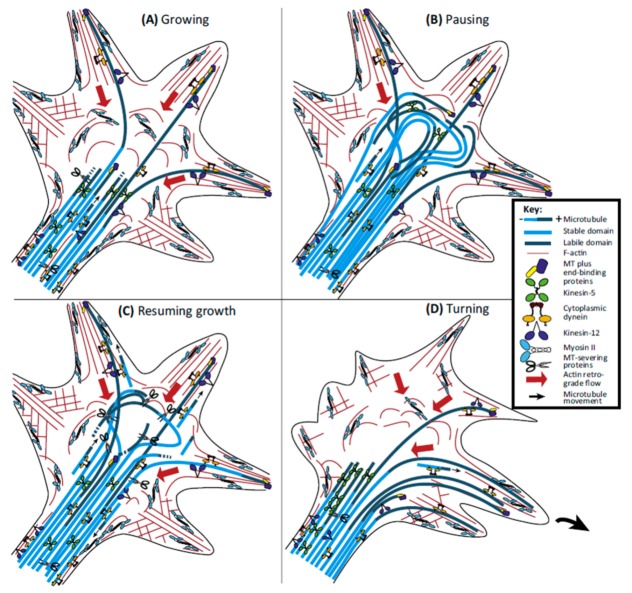
Behavior of microtubules during the different phases of the activity of the growth cone. (**A**) During the advance of the growth cone, dynein generates traction forces that allow microtubules to overcome retrograde flow and to invade the peripheral domain and filopodia. The cut-out of the labile portion of the microtubules (dark blue) regulates the length of that portion, while cutting the stable portion of microtubules (light blue) creates short and more motile ones. (**B**) When the growth cone pauses, the progress of microtubules results in a non-compartmentalized growth cone in central and peripheral domains, which is dominated by a bundle of curved microtubules. (**C**) When the axon restarts the growth, microtubules disperse rapidly and increase their cutting rate, causing shorter microtubules to be thrown away from the beam. (**D**) When the growth cone needs to turn, the forces generated by the motor proteins kinesin-5 and kinesin-12 become polarized towards the opposite side to the direction of rotation; in this way, the forces driven by dynein allow the invasion of microtubules only on the side of the growth cone that is going to turn. The small black arrows indicate the direction of the movement of short microtubules. The long red arrows indicate the retrograde flow of actin. Reprinted from: [225], with the permission of Elsevier.

**Table 1 cells-09-00358-t001:** Classification of intermediate filaments.

Intermediate Filament	Protein Name	Gene Name	Uniprot ID
Type I and II	Acidic and basic keratins	44 genes	
Type III	Desmin	*DES*	P17661
	Glial fibrillary acidic protein	*GFAP*	P14136
	Peripherin	*PRPH*	P41219
	Vimentin	*VIM*	P08670
Type IV	Internexin neuronal intermediate filament protein, alpha	*INA*	Q16352
	Neurofilament light polypeptide	*NEFL*	P07196
	Neurofilament medium polypeptide (neurofilament 3)	*NEFM*	P07197
	Neurofilament heavy polypeptide	*NEFH*	P12036
	Syncoilin 1	*SYNC*	Q9H7C4
Type V	Lamin A/C	*LMNA*	P02545
	Lamin B	*LMNB1*	P20700
		*LMNB2*	Q03252
Type VI	Nestin	*NES*	P48681
	Synemin	*SYNM*	O15061
Others	Filensin	*BFSP1*	Q12934
	Phakinin	*BFSP2*	Q13515

**Table 2 cells-09-00358-t002:** Cytoskeletal proteins and their mutations associated with neurological diseases as primary causes.

Protein (UniProt Code)/Gene Name	Mutation	Disease Name	A Brief Description of the Observations	References
Profilin (P07737)/*PFN1*	p.C71G, p.M114T, p.G118V, p.E118V, p.A20T, p.Q139L, p.T109M	Familial Amyotrophic Lateral Sclerosis (fALS)/OMIM # 105400	Mutations in PFN-1 form insoluble ubiquitinated agregates and show defects in growth cone morphology with altered actin levels.	[230,231]
	p.R136W, p.E117G, p.G15G	Sporadic Amyotrophic Lateral Sclerosis (sALS)/OMIM # 105400		[231,232,233]
β-III-Spectrin (O15020)/*SPTBN2*	p.L253P	Spinocerebellar ataxia type 5 (SCA5)/OMIM# 600224	The L253P mutant β-spectrin induces high-affinity binding that decreases spectrin-actin dynamics and consequently may alter the growing or remodelling of dendritic branches and spines.	[234]
Tubulin beta-2A chain (Q13885)/*TUBB2A*	p.D417N	TUBB2A progressive spastic ataxia syndrome	The D417N produces impaired binding to KIF1A, a neuron-specific kinesin required for transport of synaptic vesicle precursors.	[235]
Tubulin beta-4B chain (P04350)/*TUBB4AB*	p.R391H, p.R391C	Leber congenital amaurosis (LCA) with early-onset deafness/OMIM# 61789	The mutations in TUBB4B altered the dynamics of growing MTs.	[236]
Tubulin alfa-1A chain (Q71U36)/*TUBA1A*	Point mutations in the TUBA1A gene.	Lissencephaly-3/OMIM#611603	Mutations in Tubulin alfa-1A chain involve neuron migration defect, as a result, lead to brain malformations	[237]
Microtubule-associated protein tau (P10636)/*MAPT*	More than 40 pathological mutations	Primary tauopathies/OMIM # 157140	Post-translational modifications or unbalance of tau isoforms leading to intracellular inclusions in neurons and/or glia, promoting degeneration	[163]
Neurofilament light (P07196)/*NEFL*	p.P8R, p.P22S, p.N98S, p.E396K	Charcot-Marie-Tooth disease type 1F (CMT1F)/OMIM# 607734; Type 2E (CMT2E)/OMIM# 607684	A large number of NEFL mutations, localize along the gene, have been reported. Nevertheless, exist four common mutations clustered in three mutational hotspots. The pathogenic mechanism is different for each mutation in NFEL gene. The axonal maintenance is compromised cause to the neurofilament network, and axonal transport is altered.	[238]
Inverted formin-2 (Q27J81)/*INF2*	p.L57P, p.L77R, p.L69_S72del, p.C104R, p.C104W, p.C104F, p.R106P, p.G114D, p.L128P, p.L132R, p.L165P, p.E184K	Charcot-Marie-Tooth Neuropathy with Focal Segmental Glomerulosclerosis (CMTDIE)/OMIM# 614455	Mutations in INF2 dysregulates actin-dependent processes, interfering with myelinitation and mitochondrial dynamics.	[239,240,241,242]
Dystonin (Q03001)/*DST*	p.A203E; p.K4330Ter; p.E4955*; p.R206W; p.K229fs*21	Hereditary sensory autonomic neuropathy type VI (HSANVI)/OMIM# 614653	Variants of dystonin lead to abnormal actin cytoskeleton organization in fibroblast of patiens entails an alteration of cell adhesion and migration.	[204,243,244]
MACF1(Q9UPN3)/*MACF1*	p.C7135F; p.D7186Y; p.C7188F; p.C7188G; p.G6664R	Lissencephaly-9/OMIM#618325	MACF1, depending of variants cause defects in neuronal migration or axonal pathfinding or both.	[245]
Plectin (Q15149)/*PLEC*	Point mutations in the PLEC gene.	Epidermolysis bullosa simplex with muscular dystrophy/OMIM#226670	Patients shown lost of myelin of intramuscular nerves and signs of cerebellar and cerebral atrophy have been described.	[246,247]

**Table 3 cells-09-00358-t003:** Neurological disorders in which abnormalities of the cytoskeleton appear as a secondary pathophysiological mechanism.

Disease Name	Protein (UniProt Code)/Gene Name	Mutation	A brief Description of the Observations	Reference
Friedreich’s ataxia (FRDA)/OMIM #229300	Frataxin (Q16595)/FXN	GAA-triplet repeats expansion or point mutations in the *FXN* gene.	The lack of frataxin produces a reduced spreading of vimentin, increment glutathionylation of actin and MT, increment of tyrosinated tubulin and irregular distribution of phosphorylated NF-H	[248,249,250,251]
Forms of Charcot-Marie-Tooth (CMT) neuropathy: Axonal recessive (AR-CMT2-OMIM#607706#608340), axonal dominant (CMT2K-OMIM#607831) and demyelinating recessive (CMT4A-OMIM#214400)	Ganglioside-induced differentiation-associated protein 1 (Q8TB36)/*GDAP1*	Point mutations in the *GDAP1* gene.	Decreased acetylation in α-tubulin.	[252]
Axonal dominant (CMT2F-OMIM#606595) form of Charcot-Marie-Tooth (CMT) neuropathy.	Heat shock protein beta-1 (P04792)/HSPB1	p.S135F	Decreased acetylation in α-tubulin.	[253]
Sporadic Alzheimer Disease (sAD)	N/A *	Sporadic Alzheimer disease (SAD) has a variety of initiating factors.	Accumulation of intracellular neurofibrillary tangles (NFT) and extracellular amyloid plaques that accumulate in vulnerable brain regions. The hyperphosphorylation of tau and other post-translational modifications (polyglutamylation, tyrosination and detyrosination) results in microtubule destabilization and cytoskeleton abnormalities, such as actin rods are related with the axonal degeneration underlying the pathophysiology of the AD.	[254,255,256,257,258]
Familial Alzheimer Disease (fAD)/OMIM #104300	Amyloid B-protein (P05067)/*APP*; Presenilin-1(P49768)/*PSEN1*; Presenilin-2 (P49810)/*PSEN2*;	Familial Alzheimer Disease (fAD) is a genetically heterogeneous disorder.		
Sporadic Parkinson Disease (sPD)	N/A *	Sporadic Parkinson Disease (SAD) has a variety of initiating factors.	Reduced microtubule stability, mass and imbalance in the pattern of tubulin post-translational modifications and associated proteins. This aberrant stability results in deregulation of axonal transport, including trafficking of mitochondria in neurons.	[259,260,261,262,263,264]
Familial Parkinson Disease (fPD)	27 causative genes associated with PD	Familial Parkinson Disease (fAD) is a genetically heterogeneous disorder.		
Huntington’s disease (HD)/ OMIM #143100	Huntingtin (P43858)/*HTT*	Expansion of CAG repeats in the *HTT* gene	Alternative splicing is impaired in HD, altering microtubule-associated protein such as TAU and MAP2.	[265]
Spinocerebellar ataxia type 3 (SCA3)/Machado-Joseph Disease/OMIM#109150	Ataxin-3 (P54252)/*ATXN3*	Expanded CAG repeats	Expanded PolyQ protein disrupts a neuronal actin cytoskeleton.	[266]
Creutzfeldt-Jakob disease (CJD)/OMIM#123400	Major prion protein (PrP) (P04156)/*PRNP*	p.P102L, p.V180I, p.E200K	Tau and NF-L concentrations are increased in the plasma of CJD patients. Synaptic abnormalities and Cofilin phosphorylation upregulated in the terminal stage of the disease.	[267,268]
Lowe syndrome/OMIM#309000	Oculo-cerebro-renal syndrome Lowe 1(OCRL)(Q01968)/*OCRL1*	Point mutations or deletions in the *OCRL1* gene.	Abnormal F-actin dynamics in interphase cells affect endocytic recycling.	[267,268]
Lissencephaly-1/OMIM#607432	Lissencephaly-1 (LIS1)(P43034)/*PAFAH1B1*	Point mutations or deletions in the *PAFAH1B1* gene.	Reduced Lis-1 expression in mice model of Lissencephaly 1 shows severe neuronal migration defects.	[237]

* N/A, non-applicable; no proteins are listed since this is a multifactorial disease.

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
