# Peer review of "Much More Than a Scaffold: Cytoskeletal Proteins in Neurological Disorders"

_cells, 2020, doi:10.3390/cells9020358_

Round 1

Reviewer 1 Report

Munoz-Lasso et al. have authored a review article with the objective to describe and discuss the involvement of cytoskeletal proteins in neurological disorders. In general this is an important topic since practically all neurological diseases exhibit pathologic changes in the organization of their cytoskeletal components. This is clearly the case for amyotrophic lateral sclerosis, which is characterized by early defects in axonal transport and it is also obvious for tauopathies, in which a cytoskeletal protein forms insoluble aggregates. However it is less clear, whether the cytoskeletal changes are primary or secondary during the disease process. Unfortunately, the article hardly discusses these important questions. Instead it provides a lengthy introduction into the assembly of the different cytoskeletal elements, where often other review articles are cited and only the last seven pages of the whole article touch the main topic as it is promised by the title, the involvement of cytoskeletal proteins in neurological disorders. I would suggest rewriting the article and providing a clear focus on the involvement of the cytoskeleton in neurological diseases. Currently, the information appears to be gathered somehow at random.

In addition to this, several other points need attention:

It is not evident for me, why the respective proteins have been selected for Table 2. The legend says that it provides a list of mutations in cytoskeletal proteins causing neurological diseases. However, neither heat shock proteins nor amyloid beta are cytoskeletal proteins. The same is true for the term “multifactorial disease”, which is included in the protein list. In contrast, mutations in tau protein, which clearly have a causative role for most cases of FTDP-17 (a tauopathy) have not been included. Several events are mentioned, where the function of cytosketal proteins are required (line 87-93): axon growth and guidance, homeostasis, and regeneration of peripheral neurons. An important aspect is the contribution of the cytoskeleton to neuronal plasticity, which provides the basis for the adaptation of the brain to a changing environment and which is vital during the whole life of an organism. This crucial aspect is not even mentioned. Why does the description of the filament systems start with the intermediate filaments and begins with the sentence “Intermediate filaments are the third major filament system” (line 100)? At a later point the chapter on microtubules starts with the sentence: “Microtubules are the third major cytoskeletal polymers” (line 392). Figure 2 claims (according to the legend) to show the tertiary structure of intermediate filaments. This is not the case: it just shows a schematic representation of the domains and some posttranslational modifications. “GFAP is the major intermediate filament protein in Schwann cells” (line 138). This is not correct: GFAP is mainly expressed in astrocytes of the central nervous system. It is mentioned that O-GlcNAcylation is abundant in cytosketal proteins (lines 196 ff.) It should be mentioned that phosphorylation and O-glycosylation are often reciprocally correlated (since many of the same sites can be modified by both PTMs), which provides an import link to the hyperphosphorlyation of cytoskeletal proteins, which is observed in some neurodegenerative diseases (e.g., tauopathies). Different Tau isoforms are described in line 432-434. The A, B, C nomenclature is not very common and should be replaced by a nomenclature which mentions the different alternatively spliced exons (1N, 2N, 3R, 4R). It is mentioned that actin rings are present as a structure under the axonal plasma membrane (line 616). It needs to be mentioned that actin rings have also been described in dendrites (e.g., Han et al (2017) PNAS).  

Author Response

We are very grateful to the reviewer for their time and positive suggestions. The reviewer’s comments are shown in black and our replies are shown in red.

Point 1.Munoz-Lasso et al. have authored a review article with the objective to describe and discuss the involvement of cytoskeletal proteins in neurological disorders. In general this is an important topic since practically all neurological diseases exhibit pathologic changes in the organization of their cytoskeletal components. This is clearly the case for amyotrophic lateral sclerosis, which is characterized by early defects in axonal transport and it is also obvious for tauopathies, in which a cytoskeletal protein forms insoluble aggregates. However it is less clear, whether the cytoskeletal changes are primary or secondary during the disease process. Unfortunately, the article hardly discusses these important questions. Instead it provides a lengthy introduction into the assembly of the different cytoskeletal elements, where often other review articles are cited and only the last seven pages of the whole article touch the main topic as it is promised by the title, the involvement of cytoskeletal proteins in neurological disorders. I would suggest rewriting the article and providing a clear focus on the involvement of the cytoskeleton in neurological diseases. Currently, the information appears to be gathered somehow at random.

Response 1.Wethank the reviewer for his/her appreciations and comments on the structure and focus of the manuscript, and we agree with the idea thatit is very important to consider if cytoskeletal changes are primary or secondary in the disease process. Following the suggestions of the reviewer we have included in the new version of the manuscript several statements emphasizing, in any case, the primary or secondary role of cytoskeletal alterations in each of the described pathologies throughout the text. In general, in many neurodevelopmental pathologies, usually hereditary, the underlying genetic alterations directly affect cytoskeletal proteins, or proteins tightly linked to cytoskeleton function, and thus theyare a primary cause in the physiopathology of the disease. On theother hand, cytoskeletal changes are secondary mechanisms in otherneurodegenerative disorders, sometimesassociated to neuroinflammatory processes, epigenetic alterations, etc. We have tried to explain and address these issues when talking about the different pathologies throughout our review. The additions regarding this particular issues are in section 3 and conclusions.

Point 2.It is not evident for me, why the respective proteins have been selected for Table 2. The legend says that it provides a list of mutations in cytoskeletal proteins causing neurological diseases. However, neither heat shock proteins nor amyloid beta are cytoskeletal proteins. The same is true for the term “multifactorial disease”, which is included in the protein list. In contrast, mutations in tau protein, which clearly have a causative role for most cases of FTDP-17 (a tauopathy) have not been included.

Response 2.In line with the previous comment, and in accordance with the changes in the general structure of the manuscript related to the emphasis on primary or secondary mechanisms of disease, we have renamed and reorganized tables 2 and 3, so each of them clearly shows either mutations in cytoskeletal proteins that lead to neurological diseases as a primary cause (table 2) and a list of diseases in which cytoskeletal abnormalities appear as a secondary pathological mechanism (table 3). The new names for these tables are as follows:

Table 2: Cytoskeletal proteins and their mutations associated with neurological diseases as primary causes.

Table 3: Neurological disorders in which abnormalities of cytoskeleton appear as a secondary pathophysiological mechanism

The reviewer is right that, in this regard, the list of pathologies in which cytoskeleton is altered as a secondary process is extremely large, and hence, we have not been exhaustive in the list presented in table 3; given the scope of our work, we have chosen to focus on some of the most studied as well as in some very rare diseases which are of interest to us because of our research background. Nonetheless, it is true that tauopathies should have been more thoroughly reviewed, as insightfully suggested by the reviewer, so we have completed table 3 with new information now also present in the main text regarding tauopathies (see text lines 362-369 in secion 1 and 638-650 in section 3, and new Table 3). 

Point 3. Several events are mentioned, where the function of cytosketal proteins are required (line 87-93): axon growth and guidance, homeostasis, and regeneration of peripheral neurons. An important aspect is the contribution of the cytoskeleton to neuronal plasticity, which provides the basis for the adaptation of the brain to a changing environment and which is vital during the whole life of an organism. This crucial aspect is not even mentioned.

Response 3. We do understand the importance to mention the contribution of the cytoskeleton to neuronal plasticity and therefore we have mentioned it in the introduction. There is abundant available information explaining the contribution of cytoskeletal proteins to neuronal plasticity, which could even lead to another review. Therefore, we consider that this falls beyond the scope of the present work, which main goal is to give the readers a general view of all the key information related with cytoskeletal proteins in neurological diseases. Nonetheless, taking into account the interesting suggestion of the reviewer, we have cited some excellent, recent and more detailed reviews on the topic ofneuronal plasticity (see text lines 83)

Point 4.Why does the description of the filament systems start with the intermediate filaments and begins with the sentence “Intermediate filaments are the third major filament system” (line 100)? At a later point the chapter on microtubules starts with the sentence: “Microtubules are the third major cytoskeletal polymers” (line 392).

Response 4.We appreciate the correction and consequently we have changed the description of intermediate filaments in section 1.1 and microtubules in section 1.3. (see text lines 93 and 329)

Point 5.Figure 2 claims (according to the legend) to show the tertiary structure of intermediate filaments. This is not the case: it just shows a schematic representation of the domains and some posttranslational modifications. “GFAP is the major intermediate filament protein in Schwann cells” (line 138). This is not correct: GFAP is mainly expressed in astrocytes of the central nervous system.

Response 5.We appreciate both insightful comments. Consequently, we have corrected the title of Figure 2 as suggested by the reviewer, and lines 116-117 to clarify the information provided. It is also is true that GFAP shows a selective cytoplasmic expression of astrocytes. We did not want to affirm that GFAP was the major intermediate filament protein in Schwann cells upon Astrocytes, what we wanted to communicate was that GFAP presents a high expression in Schwann cells, and have changed the sentence as indicated.   (see text lines 116-117)

Point 6.It is mentioned that O-GlcNAcylation is abundant in cytosketal proteins (lines 196 ff.) It should be mentioned that phosphorylation and O-glycosylation are often reciprocally correlated (since many of the same sites can be modified by both PTMs), which provides an import link to the hyperphosphorlyation of cytoskeletal proteins, which is observed in some neurodegenerative diseases (e.g., tauopathies).

Response 6.We agree with the reviewer in the importance to mention the reciprocal relationship between this two posttranslational modifications, and therefore we address the reader to an interesting review that explains this topic in great detail in the section 3.2. (see text lines 703-705)

Point 7. Different Tau isoforms are described in line 432-434. The A, B, C nomenclature is not very common and should be replaced by a nomenclature which mentions the different alternatively spliced exons (1N, 2N, 3R, 4R).

Response 7.We have followed the recommendation of the reviewer and changed the nomenclature of the nine isoforms of tau by using the nomenclature available in UNIPROT that mentions the different alternatively spliced exons. We have not found options for tau A and tau G, therefore we have decided to leave it with the nomenclature available in UNIPROT. (see text lines 355-359)

Point 8.It is mentioned that actin rings are present as a structure under the axonal plasma membrane (line 616). It needs to be mentioned that actin rings have also been described in dendrites (e.g., Han et al (2017) PNAS).  

Response 8.We appreciate the recommendation of the reviewer and therefore we have added the reference of the work that shows that actin rings have been found in dendrites. (see text lines 532)

Reviewer 2 Report

The review by Muñoz-Lasso et al. focuses on an interesting topic, cytoskeletal proteins and their involvement in different neurological disorders. 

This review provides balanced assessment of the three main types of filaments: actin filaments, microtubules, and intermediate filaments (neurofilaments). The article is quite extensive and highlights important data.

Nonetheless, I have a suggestion which would further improve the paper. One chapter dedicated to the cytolinker proteins (e.g. plakin family of cyoskeletal crosslinkers, such as plectin) would be a welcome addition. Especially since it would further emphasize the interdependent nature of the three types of filaments.

Author Response

We are very grateful to the reviewer for their time and positive suggestions. The reviewer’s comments are shown in black and our replies are shown in red.

Point 1.The review by Muñoz-Lassoet al. focuses on an interesting topic, cytoskeletal proteins and their involvement in different neurological disorders. 

This review provides balanced assessment of the three main types of filaments: actin filaments, microtubules, and intermediate filaments (neurofilaments). The article is quite extensive and highlights important data.

Nonetheless, I have a suggestion which would further improve the paper. One chapter dedicated to the cytolinker proteins (e.g. plakinfamily of cyoskeletal crosslinkers, such as plectin) would be a welcome addition. Especially since it would further emphasize the interdependent nature of the three types of filaments.

Response 1.We thank the reviewer for his/her kind comments and appraisal of our manuscript. Following his/her recommendation, we have included a new section (1.4) in order to offer an overview of the role of cytolinker proteins, which we think has enriched the manuscript; besides, we have also included specific information on some representative examples of the involvement of mutations in plakins regarding neurological disorders (see text lines 431-457 in section 1; text lines 662-670 and 764-773 in section 3; and Table 2).

Round 2

Reviewer 1 Report

Some limitations of the original submission have been sufficiently addressed in the revised version of the manuscript by Munoz-Lasso et al. The authors have included some information whether cytoskeletal changes are primary or secondary during the disease process; a statement with respect to the role of the cytoskeleton in neuronal plasticity has been added; the legend of Figure 2 has been adjusted to describe what has actually been shown; the nomenclature of tau has been adjusted and information on GFAP with respect to its expression in astrocytes has been included. And, importantly, Table 2 and 3 are much clearer in the revised version.

I am still not convinced by the structure of the article, which starts with a lengthy general discussion, while the main focus, as it is promised in the title, is treated only starting from page 22. Many textbooks or more comprehensive review articles provide a better and more complete general introduction than this part of the article. I also find it somehow strange that figures from other published review articles have been used (e.g., Fig. 3 and Fig. 8). However, this may be a matter of taste.

Some aspects still require attention:

I still don’t get the point, why the description of the filament systems starts with the intermediate filaments and begins with the sentence “Intermediate filaments are the third main polymer system” (line 75). Why not simply starting with the first or remove the numbering at all. The legend of Figure 3 (line 127) states: “self-assembly of filaments” – should be rewritten to “self-assembly of neurofilaments”. Microtubules in the axon are not continuous but present as an discontinuous array (line 48) “The shorter isoforms of tau are thought to stabilize MTs while the longer allow plasticity of the cytoskeleton” (line 426/427) – there is no evidence that tau stabilizes microtubules from tau KO or acute tau knockdown experiments. I also do not get the point why shorter tau isoforms (with less MT-binding regions) should stabilize MTs more; what is the reference for this.

Author Response

We are very grateful again to the reviewer for their time and new suggestions. The reviewer’s comments are shown in black and our replies are shown in red.

I am still not convinced by the structure of the article, which starts with a lengthy general discussion, while the main focus, as it is promised in the title, is treated only starting from page 22. Many textbooks or more comprehensive review articles provide a better and more complete general introduction than this part of the article. I also find it somehow strange that figures from other published review articles have been used (e.g., Fig. 3 and Fig. 8). However, this may be a matter of taste.

Following the recommendations of the reviewer we have replaced figures 3 and 8 with original figures, which now are more in accordance with the content of the manuscript.

Some aspects still require attention:

Point 1.I still don’t get the point, why the description of the filament systems starts with the intermediate filaments and begins with the sentence “Intermediate filaments are the third main polymer system” (line 75). Why not simply starting with the first or remove the numbering at all.

Response 1.We appreciate the correction and consequently we have totally removed any reference to the numbering (see text lines 93, 164 and 325).

Point 2.The legend of Figure 3 (line 127) states: “self-assembly of filaments” – should be rewritten to “self-assembly of neurofilaments”.

Response 2.As we explained above, since figure 3 has been re-elaborated, we have accordingly written a new figure legend and incorporated the title suggested by the reviewer.

Point 3.Microtubules in the axon are not continuous but present as an discontinuous array (line 48)

Response 3.We thank the reviewer for the insightful correction, and we have re-written the corresponding sentence in the text (see line 75).

Point 4.“The shorter isoforms of tau are thought to stabilize MTs while the longer allow plasticity of the cytoskeleton” (line 426/427) – there is no evidence that tau stabilizes microtubules from tau KO or acute tau knockdown experiments. I also do not get the point why shorter tau isoforms (with less MT-binding regions) should stabilize MTs more; what is the reference for this

Response 4.The reviewer is right that the sentence is not accurate, so we have directly removed it from the manuscript and thank the reviewer for noticing.